# NOISE ROBUST GRAPH LEARNING UNDER FEATURE-DEPENDENT GRAPH-NOISE

## ABSTRACT

In real-world scenarios, node features frequently exhibit noise due to various factors, making GNNs vulnerable. Various methods enhance robustness, but they make an unrealistic assumption that the noise in node features is independent of the graph structure and node labels, restricting their practicality. To this end, we introduce a more realistic noise scenario, called feature-dependent graph-noise (FDGN), where noisy node features may entail both structure and label noise, and propose a deep generative model that directly captures the causal relationships among the variables in the DGP of FDGN. We formulate a tractable and feasible learning objective based on variational inference and provide detailed discussions on the instantiations of the model components corresponding to the derived terms. Our proposed method, PRINGLE, outperforms baselines on commonly used benchmark datasets and newly introduced real-world graph datasets that simulate FDGN in e-commerce systems. Our code is available at https://anonymous.4open.science/r/FDGN-PRINGLE-4B31/

## 1 INTRODUCTION

Graphs are prevalent data structure that exist across diverse domains, including chemistry, economics, and social media. Given their pervasive presence, it becomes essential to obtain effective graph representations. In recent years, graph neural networks (GNNs) have demonstrated remarkable achievements in graph representation learning and have been extensively applied in numerous downstream tasks (Yao et al., 2019; Wang et al., 2019a; Wu & Hooi, 2023).

However, in the majority of real-world scenarios, node features frequently exhibit noise due to various factors, leading to the creation of inaccurate graph representations (Liu et al., 2021; Jin et al., 2022). For instance, in social networks, users may create fake profiles or posting, resulting in noisy node features. Similarly, in product co-purchase networks found in e-commerce systems, node features can be contaminated by fake reviews. Recent studies have revealed the vulnerability of GNNs to such scenarios, highlighting the necessity to design robust GNN models against noisy node features. To this end, various methods have been proposed to make a huge success in terms of model robustness (Liu et al., 2021; Jin et al., 2022).

While such existing robust GNN models have proven effective, we argue that their practicality is restricted by an unrealistic assumption regarding graph noise: they assume that the noise in node features is independent of the graph structure or node labels. For example, in the conventional graph noise (CGN) assumption in terms of node features (Fig. 1(b)), Bob's fake profile does not influence other nodes, which is also explained by the data generating process (DGP) of CGN (See Fig. 2(a)) in which no causal relation-

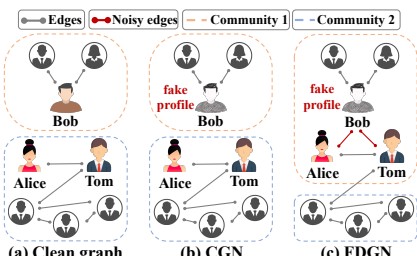

(a) Clean graph    (b) CGN    (c) FDGN

Figure 1: Social network examples.

ships exist among the noisy node features $X$, graph structure $A$, and node labels $Y$. However, in reality (See Fig. 1(c)), other users may make connections with Bob based on his fake profile (i.e., structure noise), which may also eventually change their community (i.e., label noise), and such causal relationships among $X$, $A$, and $Y$ (i.e., $A \leftarrow X$, $Y \leftarrow X$, and $Y \leftarrow A$) are depicted in Fig. 2(b).

As another real-world scenario, consider a product co-purchase network in an e-commerce platform, where nodes denote products and edges denote the co-purchase relationship between products.

In this case, fake reviews on products written by a fraudster would make other users purchase irrelevant products, which adds irrelevant edges between products (i.e., structure noise). Consequently, this would make the automated product category labeling system to inaccurately annotate product categories (i.e., label noise), as it relies on the node features and the graph structure, both of which are contaminated.

These examples demonstrate that, in reality, *noisy node features may entail both structure and label noise*. To reflect such a realistic scenario in graph learning, we define **f**eature-**d**ependent **g**raph-**n**oise (FDGN) and model the data generation process in a graphical model. We also observe that existing robust GNN models indeed fail to generalize effectively in a more realistic FDGN due to their inadequate assumption on the data generation process (Sec 3.2).

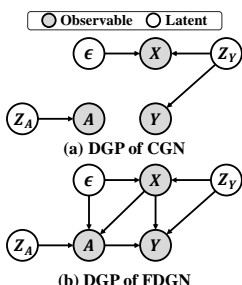

Figure 2: A directed graphical model indicating a DGP of (a) CGN, and (b) FDGN.

In this work, we propose a **pri**ncipled **n**oisy **g**raph **le**arning framework (PRINGLE), which operates under a more realistic FDGN assumption. We first define the DGP of FDGN as shown in Fig. 2(b). More precisely, we introduce three observable variables (i.e., node features $X$, graph structure $A$, and node labels $Y$) and three latent variables (i.e., noise incurring variable $\epsilon$, latent clean graph structure $Z_A$, and latent clean node labels $Z_Y$), while defining causal relationships among these variables to represent the data generation process of FDGN. We then devise a deep generative model that directly captures the causal relationships among the variables in the DGP of FDGN, and derive a tractable and feasible learning objective based on variational inference. It is worth noting that although the main focus of this paper is on the node feature noise and its influence across the graph, our proposed robust graph learning framework is capable of generalizing not only to feature-dependent graph noise (FDGN), but also to independent structure/feature/label noise that is also prevalent in real-world applications. This implies that PRINGLE has a wider range of applicability than existing robust GNN models. In order to conduct a rigorous evaluation of PRINGLE, we perform evaluations on not only existing benchmark datasets, but also newly introduced real-world graph datasets that simulate FDGN within e-commerce systems, providing a valuable alternative to synthetic settings.

In summary, the main contributions of the paper are three-fold:

- We investigate limitations of the conventional graph noise assumption in terms of node features, and introduce a more realistic graph noise scenario, **f**eature-**d**ependent **g**raph-**n**oise (FDGN). To the best of our knowledge, this is the first attempt to understand the data generation process in graph domain that mimics the noise scenario in real-world.

- We propose the **pri**ncipled **n**oisy **g**raph **le**arning framework (PRINGLE) which addresses FDGN by modeling its DGP. PRINGLE outperforms state-of-the-art baselines in node classification and link prediction tasks under various scenarios including feature-dependent graph-noise and independent structure/feature/label noise.

- In addition to existing benchmark datasets in which noise is synthetically generated, we further introduce novel graph benchmark datasets that simulate FDGN within e-commerce systems, which is expected to foster practical research in noise-robust graph learning.

## 2    RELATED WORKS

The objective of noise-robust graph learning is to train GNN models when the input graph data exhibits one or more of the following types of noise: 1) node feature noise, 2) graph structure noise, and/or 3) node label noise. The majority of existing approaches focus primarily on graphs containing only a single type of noise.

**Feature noise-robust graph learning.**   To address the noisy node features, various approaches including adversarial training (Tian et al., 2023) and test-time graph transformation (Jin et al., 2022), have been proposed. Additionally, recent studies have highlighted the significance of fully leveraging structural information. AirGNN (Liu et al., 2021) proposed a novel message passing mechanism that identifies the nodes with noisy features and learns node-wise adaptive coefficients that balance the feature aggregation and use of their own noisy features. The identification process is guided by the intuition that nodes with noisy features tend to have dissimilar features within their local neighborhoods. In summary, this approach tackles the noisy node features while assuming that the structure of the input graph is noise-free.

**Structure noise-robust graph learning.** To address the noisy graph structure, various approaches including robust message passing scheme (Lei et al., 2022) and graph structure learning (Jin et al., 2020) have been proposed. Among them, a representative approach is based on the graph structure learning (GSL), which aims to learn a refined graph structure from a given graph. Specifically, ProGNN (Jin et al., 2020) learns a graph structure aiming at satisfying the real-world graph properties, e.g., feature-smoothness. Moreover, RSGNN (Dai et al., 2022) aims to train a graph structure learner, which is composed of an MLP encoder and a regularizer for enhancing feature-smoothness, which encourages the nodes with similar features to be connected in the refined structure. STABLE (Li et al., 2022a) aims at acquiring robust node representations by roughly refining the given graph by removing easily detectable noisy edges, typically those connecting nodes with low feature-similarity, after which node representations are learned in an unsupervised manner. Finally, based on these representations, a kNN graph is constructed to serve as the refined structure. In summary, these methods tackle the noisy graph structure while assuming that node features are noise-free.

**Label noise-robust graph learning.** There are numerous studies that address the noisy labels on non-graph data (Li et al., 2020; Wang et al., 2019b; Yao et al., 2021). However, since they are not directly applicable for graph data (Dai et al., 2021), recent studies investigated the label noise-robust graph learning methods. The key idea of NRGNN (Dai et al., 2021) is to correct the predictions of unlabeled nodes affected by information propagation from falsely labeled nodes. To do so, NRGNN learns a new graph structure where two nodes with similar features are connected, based on the assumption that two nodes are more likely to have the same label if they have similar features. This strategy mitigates the information propagation from falsely labeled nodes. RTGNN (Qian et al., 2023) identifies clean labeled samples from noisy labeled ones based on the small-loss criteria and leverages pseudo-labeling to supplement the labeled nodes. However, nodes with noisy features or structures would yield a large-loss even if their labels do not contain any noise, which results in inaccuracies in the selection of clean labeled samples. Therefore, these methods tackle the noisy node labels while assuming that both node features and graph structure are noise-free.

**Generative graph learning.** Apart from the noise-robust graph learning methods, several studies adopted deep generative modeling to infer latent graph data (Ma et al., 2019; Elinas et al., 2020; Lao et al., 2022). Most recently, WSGNN (Lao et al., 2022) uses a probabilistic generative approach and variational inference to infer the latent graph structure and node labels. However, it assumes noise-free graphs, making it less effective in handling various noise scenarios commonly found in real-world applications.

In summary, each of the aforementioned methods assumes the completeness of at least one of the data sources, i.e., node features, graph structures, or node labels. In contrast, our proposed method is constructed under a more realistic FDGN assumption, where noise in node features may result in structural and label noise. This fundamental difference liberates the proposed method from such limited assumptions.

## 3 FEATURE-DEPENDENT GRAPH-NOISE (FDGN)

### 3.1 DEFINITION

In this section, we define a realistic graph noise assumption, i.e., FDGN, and its DGP. Specifically, we assume that the graph containing FDGN is generated according to the graphical model in Fig. 2(b). We first introduce each variable in the graphical model, and then explain each relationship between variables through two real-world applications where FDGN exists: social networks (i.e., user-user graph) and co-purchase networks (i.e., product-product graph) within e-commerce systems.

In the graphical model in Fig. 2(b), $X$ represents the node features, which may contain noisy node features; $Y$ represents the observed node labels, which may contain noisy labels; $A$ represents the observed edges between two nodes, which may contain noisy edges; $\epsilon$ represents the environment variable that causes the noise; $Z_Y$ represents the latent clean node labels; $Z_A$ represents the latent clean graph structure that contains all potential connections between nodes. We provide explanations for each relationship in the graphical model of FDGN shown in Fig. 2(b):

- $X \leftarrow (\epsilon, Z_Y)$: $\epsilon$ and $Z_Y$ are causes of $X$. In social networks, users create their profiles and postings (i.e., $X$) regarding their true communities or interests (i.e., $Z_Y$). However, if users decide to display fake profiles for some reason (i.e., $\epsilon$), $\epsilon$ is a cause of the noisy node features $X$. In co-purchase networks, the reviews and descriptions of products (i.e., $X$) are written regarding their true categories (i.e., $Z_Y$). However, if a fraudster (i.e., $\epsilon$) writes fake reviews on products, $\epsilon$ is a cause of the noisy node features (i.e., $X$).

- $A \leftarrow (Z_A, X)$: $Z_A$ and $X$ are causes of $A$. In social networks, the follow relationship among users (i.e., $A$) are made based on their latent relationships (i.e., $Z_A$). However, if a user creates a fake profile (i.e., $X$), some irrelevant users may follow the user based on his/her fake profile, which leads to noisy edges (i.e., $A$). In co-purchase networks, the co-purchase relationship among products (i.e., $A$) are made based on their true relevance (i.e., $Z_A$). However, if a fraudster writes fake reviews (i.e., $X$) on multiple products, some irrelevant products may be connected by co-purchase relationship, which leads to noisy edges (i.e., $A$).

- $A \leftarrow \epsilon$: To provide a broader scope, we also posit that $\epsilon$ is a potential cause of $A$. This extension is well-founded, as real-world applications often exhibit graph structure noise originating from various sources in addition to the feature-dependent noise (Liu et al., 2022; Fatemi et al., 2021).

- $Y \leftarrow (Z_Y, X, A)$: $Z_Y$, $X$, and $A$ are causes of $Y$. In social networks, the true communities (or interests) of users (i.e., $Z_Y$) are leveraged to promote products to targeted users within a community (Ma et al., 2021). To detect the communities, both node features and graph structures are utilized. However, if a user has noisy node features (i.e., $X$) or noisy edges (i.e., $A$), the user may be assigned to a wrong community (or interest), which leads to noisy labels (i.e., $Y$). In co-purchase networks, machine learning-based automated labeling techniques are widely used in e-commerce systems to label the true categories of products (i.e., $Z_Y$) since new products are continuously released. However, the automated labeling systems may become inaccurate due to noisy node features (i.e., $X$) and noisy graph structures (i.e., $A$), which leads to noisy node labels (i.e., $Y$).

For simplicity, we assume that $\epsilon$ is not a cause of $Y$. This assumption aligns with practical scenarios in real-world applications, where an instance is more likely to be mislabeled due to confusing or noisy features rather than arbitrary sources, i.e., instance-dependent label-noise (Yao et al., 2021; Berthon et al., 2021). In other words, label noise in graphs is predominantly caused by confusing or noisy features and graph structure, i.e., $Y \leftarrow (X, A)$, rather than an arbitrary external factor, i.e., $Y \nleftarrow \epsilon$.

## 3.2 PRELIMINARY ANALYSIS ON FDGN

We conduct an analysis to examine how well existing robust GNN models generalize to FDGN. We generate three types of noise: random feature noise (Liu et al., 2021), random structure noise (Li et al., 2022a), and random label noise (Dai et al., 2021; Qian et al., 2023), following

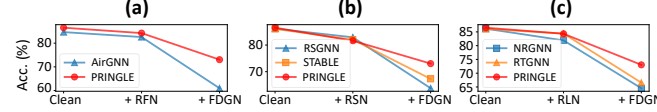

Figure 3: Node classification performance of baselines and PRINGLE on Cora dataset. Here, + *RFN*, + *RSN*, and + *RLN* indicate injecting the random feature noise, random structure noise, and random label noise into the original graph, respectively.

the convention of the existing studies. As baselines, we consider (a) AirGNN as a feature noise-robust graph learning method, (b) RSGNN and STABLE as structure noise-robust graph learning methods, and (c) NRGNN and RTGNN as label noise-robust graph learning methods. A comprehensive description of each method is provided in Sec. 2.

We observe that while existing noise-robust graph learning methods perform well under the random feature noise (i.e., + RFN in Fig. 3(a)), structure noise (+ RSN in Fig. 3(b)), and label noise (+ RLN in Fig. 3(c)), their performance significantly drops under FDGN (i.e., + FDGN in Fig. 3). In contrast, our proposed method (PRINGLE) demonstrates competitive results under the random noise scenarios, while notably outperforming the baselines under FDGN. This points to a key distinction – existing noise-robust graph learning methods struggle to generalize to FDGN due to their limited assumptions regarding the graph noise. Specifically, as summarized in Sec. 2, each category of methods assumes at least one of the data sources is noise-free: node features, graph structures, or node labels. Nevertheless, the causal relationships among $X$, $A$, and $Y$ within the data generation process of FDGN gives rise to scenarios involving concurrent feature, structure, and label noise. Consequently, existing robust GNN models fall short of effectively generalizing to FDGN, as they overlook such underlying relationships among noise types, leading to model designs assuming the completeness of at least one data source. Conversely, PRINGLE directly captures the underlying relationships by modeling the DGP of FDGN, resulting in superior generalization to FDGN.

## 4 PRINCIPLED NOISY GRAPH LEARNING FRAMEWORK (PRINGLE)

In this section, we propose a principled noisy graph learning framework (PRINGLE) to tackle more realistic noise scenario, FDGN. It is essential to highlight that under FDGN, noisy node features entail both structure and label noise, resulting in a graph that does not contain any noise-free data sources,

i.e., a graph with noisy $X$, noisy $A$, and noisy $Y$. This point presents a non-trivial challenge for the existing noise-robust graph learning methods to tackle FDGN, as they assume the completeness of at least one data source. To address this challenge, we design a deep generative model that directly models the DGP of FDGN, thereby capturing the causal relationships among the variables that introduce noise. First, we derive the Evidence Lower Bound (ELBO) for the observed data log-likelihood $P(X, A, Y)$ based on the graphical model of FDGN (**Section 4.2**). Subsequently, we discuss model instantiations for the model components corresponding to the derived terms, including implementation details (**Section 4.3**). It is essential to highlight that our approach can handle both node classification and link prediction tasks, making it versatile and applicable in various situations. Appendix A shows the overall architecture and training algorithm of PRINGLE.

### 4.1 PROBLEM STATEMENT

**Notations** We have an undirected and unweighted graph $\mathcal{G} = \langle \mathcal{V}, \mathcal{E} \rangle$ where $\mathcal{V} = \{v_1, ..., v_N\}$ represents the set of nodes and $\mathcal{E} \in \mathcal{V} \times \mathcal{V}$ indicates the set of edges. Each node $v_i$ has the node features $\mathbf{X}_i \in \mathbb{R}^F$ and node labels $\mathbf{Y}_i \in \{0, 1\}^C$, where $F$ is the number of features for each node and $C$ indicates the number of classes. We represent the observed graph structure using the adjacency matrix $\mathbf{A} \in \mathbb{R}^{N \times N}$, where $\mathbf{A}_{ij} = 1$ if there is an edge connecting nodes $v_i$ and $v_j$, and $\mathbf{A}_{ij} = 0$ otherwise.

**Tasks: node classification and link prediction** In the node classification task, we assume the semi-supervised setting where only a portion of nodes are labeled (i.e., $\mathcal{V}^L$). Our objective is to predict the labels of unlabeled nodes (i.e., $\mathcal{V}^U$) by inferring the latent clean node label $Z_Y$. In the link prediction task, our goal is to predict reliable links based on partially observed edges by inferring the latent clean graph structure $Z_A$. It is important to note that, according to the FDGN assumption, the observed node features, graph structure, and node labels may contain noise.

### 4.2 MODEL FORMULATION

We commence by modeling joint distribution $P(X, A, Y)$. We assume that the joint distribution $P(X, A, Y)$ is differentiable nearly everywhere regarding both $\theta$ and the latent variables $(\epsilon, Z_A, Z_Y)$. Note that the generative parameter $\theta$ serves as the decoder network that models the distribution $P(X, A, Y)$. The joint distribution of $P(X, A, Y)$ can be represented as: $p_\theta(X, A, Y) = \int_\epsilon \int_{Z_A} \int_{Z_Y} p_\theta(X, A, Y, \epsilon, Z_A, Z_Y) d\epsilon dZ_A dZ_Y$. However, computing this evidence integral is either intractable to calculate in closed form or requires exponential time. As the evidence integral is intractable for computation, calculating the conditional distribution of latent variables $p_\theta(\epsilon, Z_A, Z_Y | X, A, Y)$ is also intractable: $p_\theta(\epsilon, Z_A, Z_Y | X, A, Y) = \frac{p_\theta(X, A, Y, \epsilon, Z_A, Z_Y)}{p_\theta(X, A, Y)}$.

To infer the latent variables, we introduce an inference network $\phi$ to model the variational distribution $q_\phi(\epsilon, Z_A, Z_Y | X, A, Y)$, which serves as an approximation to the posterior $p_\theta(\epsilon, Z_A, Z_Y | X, A, Y)$. To put it more concretely, the posterior distribution can be decomposed into three distributions determined by trainable parameters $\phi_1$, $\phi_2$, and $\phi_3$. Based on the observed conditional independence relationships [1], we decompose $q_\phi(\epsilon, Z_A, Z_Y | X, A, Y)$ as follows:

$$q_\phi(\epsilon, Z_A, Z_Y | X, A, Y) = q_{\phi_1}(Z_A | X, A, \epsilon) q_{\phi_2}(\epsilon | X, A, Z_Y) q_{\phi_3}(Z_Y | X, A, Y). \tag{1}$$

For simplicity, we introduce two additional assumptions. First, when the node features $X$ and observed graph structure $A$ are given, latent clean graph structure $Z_A$ is conditionally independent from the noise-incurring variable $\epsilon$, i.e., $q_{\phi_1}(Z_A | X, A, \epsilon) = q_{\phi_1}(Z_A | X, A)$. Second, when $X$ and $A$ are given, latent clean labels $Z_Y$ is conditionally independent from the observed node labels $Y$, i.e., $q_{\phi_3}(Z_Y | X, A, Y) = q_{\phi_3}(Z_Y | X, A)$. This approximation, known as the mean-field method, is a prevalent technique utilized in variational inference-based methods (Ma et al., 2019; Lao et al., 2022). As a result, we can simplify Eqn. 1 as follows:

$$q_\phi(\epsilon, Z_A, Z_Y | X, A, Y) = q_{\phi_1}(Z_A | X, A) q_{\phi_2}(\epsilon | X, A, Z_Y) q_{\phi_3}(Z_Y | X, A). \tag{2}$$

To jointly optimize the parameter $\phi$ and $\theta$, we adopt the variational inference framework (Blei et al., 2017; Yao et al., 2021) to optimize the Evidence Lower-BOund (ELBO) of the marginal likelihood for observed data, rather than optimizing the marginal likelihood directly. Specifically, we derive the negative ELBO, i.e., $\mathcal{L}_{\text{ELBO}}$, as follows:

$$\mathcal{L}_{\text{ELBO}} = -\mathbb{E}_{Z_A \sim q_{\phi_1}(Z_A | X, A)} \mathbb{E}_{\epsilon \sim q_{\phi_2}(\epsilon | X, A, Z_Y)} \left[ \log(p_{\theta_1}(A | X, \epsilon, Z_A)) \right] + kl(q_{\phi_1}(Z_A | X, A) || p(Z_A))$$

$$- \mathbb{E}_{\epsilon \sim q_{\phi_2}(\epsilon | X, A, Z_Y)} \mathbb{E}_{Z_Y \sim q_{\phi_3}(Z_Y | X, A)} \left[ \log(p_{\theta_2}(X | \epsilon, Z_Y)) \right] + \mathbb{E}_{Z_Y \sim q_{\phi_3}(Z_Y | X, A)} \left[ kl(q_{\phi_2}(\epsilon | X, A, Z_Y) || p(\epsilon)) \right]$$

$$- \mathbb{E}_{Z_Y \sim q_{\phi_3}(Z_Y | X, A)} \left[ \log(p_{\theta_3}(Y | X, A, Z_Y)) \right] + kl(q_{\phi_3}(Z_Y | X, A) || p(Z_Y)) \tag{3}$$

---

[1] We observe the following conditional independence relationships in Fig. 2(b): (1) $Z_A \perp Y | X, A, \epsilon$, (2) $Z_A \perp Z_Y | A, X, \epsilon$, (3) $\epsilon \perp Y | Z_Y, X, A$.

where $kl(\cdot||\cdot)$ denotes KL divergence. The derivation details are provided in Appendix B. The encoders $\phi_1$, $\phi_2$, and $\phi_3$ infer three latent variables, while $\theta_1$, $\theta_2$, and $\theta_3$ are decoder networks used for generating three observable variables. Our objective is to find the optimal values of $\phi = \{\phi_1, \phi_2, \phi_3\}$ and $\theta = \{\theta_1, \theta_2, \theta_3\}$ that minimize the value of $\mathcal{L}_{\text{ELBO}}$, expressed as $\text{argmin}_{\theta,\phi} \mathcal{L}_{\text{ELBO}}$.

## 4.3 MODEL INSTANTIATIONS

### 4.3.1 INFERENCE OF LATENT VARIABLES $Z_A$, $Z_Y$, AND $\epsilon$

**Inference of $Z_A$.** The encoder $q_{\phi_1}(Z_A|X, A)$ is instantiated as a graph structure learner predicting the probability $p_{ij}$, where we consider each edge in $Z_A$ between node $v_i$ and $v_j$ as an independent Bernoulli random variable, i.e., $\text{Bern}(p_{ij})$ (Wu et al., 2020). Specifically, we use a GCN encoder to acquire deterministic node embeddings, i.e., $\mathbf{Z} = \text{GCN}_{\phi_1}(\mathbf{X}, \mathbf{A}) \in \mathbb{R}^{N \times d_1}$, where $d_1$ is the dimension of node embedding. To acquire the latent graph $\hat{\mathbf{A}} = \{\hat{a}_{ij}\}_{N \times N}$, we sample from $\hat{a}_{ij} \sim \text{Bern}(\hat{p}_{ij})$ where the estimated parameter $\hat{p}_{ij}$ is computed as follows: $\hat{p}_{ij} = \rho(s(\mathbf{Z}_i, \mathbf{Z}_j))$, where $s(\cdot, \cdot)$ is a cosine similarity function and $\rho$ is the ReLU activation function.

**Inference of $Z_Y$.** The encoder $q_{\phi_3}(Z_Y|X, A)$ is instantiated as a GCN classifier that deterministically outputs the probability of the latent label $Z_Y$. In the practical implementation, we infer $Z_Y$ through $\hat{\mathbf{Y}} = \text{GCN}_{\phi_3}(\mathbf{X}, \hat{\mathbf{A}}) \in \mathbb{R}^{N \times C}$ because $\hat{\mathbf{A}}$ contains not only $\mathbf{A}$, but also rich structural information that is missing in $\mathbf{A}$, which helps enhance the inference of $Z_Y$. We introduce the node label classification loss $\mathcal{L}_{\text{cls-enc}} = \sum_{i \in \mathcal{V}^L} \text{CE}(\hat{\mathbf{Y}}_i, \mathbf{Y}_i)$, where CE is the cross entropy loss.

**Inference of $\epsilon$.** We decompose $q_{\phi_2}(\epsilon|X, A, Z_Y)$ into $q_{\phi_{21}}(\epsilon_X|X, Z_Y)$ and $q_{\phi_{22}}(\epsilon_A|X, A)$, where $\epsilon_X$ and $\epsilon_A$ are independent variables that incur the feature and structure noise, respectively. For the encoder $q_{\phi_{21}}(\epsilon_X|X, Z_Y)$, we use an MLP encoder that takes $CONCAT(X, Z_Y)$ as an input and outputs $\epsilon_X$. For the encoder $q_{\phi_{22}}(\epsilon_A|X, A)$, we regard $\epsilon_A$ as a set of scores indicating the likelihood of each observed edge being noisy or not. Inspired by an early-learning phenomenon (Arpit et al., 2017), we compute the set of link prediction losses using MSE on the observed edges $\mathcal{E}$ as $\{(1 - \hat{p}_{ij}^{el})^2|(i,j) \in \mathcal{E}\}$, where $\hat{p}_{ij}^{el}$ represents the $\hat{p}_{ij}$ value at the final epoch during early-learning phase. Therefore, an edge with high $\hat{p}_{ij}^{el}$ value can be considered as a clean edge. Hence, $\epsilon_A$ is instantiated as $\{\hat{p}_{ij}^{el}|(i,j) \in \mathcal{E}\}$.

### 4.3.2 LOSS TERMS IN $\mathcal{L}_{\text{ELBO}}$

**Loss term $kl(q_{\phi_1}(Z_A|X, A)||p(Z_A))$.** To minimize this term, we encourage $Z_A$ to align with our prior knowledge, i.e., $p(Z_A)$, that the latent graph structure predominantly consists of assortative edges that have the potential to enhance feature propagation within GNNs (Zhao et al., 2023). In recent studies (Choi et al., 2022; Dai et al., 2022), the $\gamma$-hop subgraph similarity has served as a potent metric for identifying assortative edges. Hence, we model $q_{\phi_1}(Z_A|X, A)$ to minimize the KL divergence between the latent graph structure $\hat{\mathbf{A}}$ and prior graph structure $\mathbf{A}^{\text{p}} = \{a_{ij}^{\text{p}}\}_{N \times N}$, where $a_{ij}^{\text{p}}$ is sampled from a Bernoulli distribution with a probability as given by the $\gamma$-hop subgraph similarity. This leads the estimated probability $\hat{p}_{ij}$ between two nodes to increase if they exhibit a high subgraph similarity. However, computing $\hat{p}_{ij}$ in every epoch is impractical for large graphs, i.e., $O(N^2)$. To mitigate the issue, we pre-define a candidate graph that consists of the observed edge set $\mathcal{E}$ and a $k$-NN graph based on the $\gamma$-hop subgraph similarity. Then, we compute the $\hat{p}_{ij}$ values of the edges in a candidate graph. Please refer to the Appendix C for detailed information on how this approach mitigates the complexity issue.

**Loss term $kl(q_{\phi_3}(Z_Y|X, A)||p(Z_Y))$.** To minimize this term, we encourage $Z_Y$ to align with our prior knowledge, i.e., $p(Z_Y)$, that the two end nodes on the latent graph structure $Z_A$ are expected to have an identical latent labels $Z_Y$, known as class homophily (McPherson et al., 2001). Hence, we model $Z_Y$ to minimize the KL divergence between the probability predictions $\hat{\mathbf{Y}}$ of each node and its first order neighbors in the estimated latent structure $\hat{\mathbf{A}}$. The implemented loss function is as: $\mathcal{L}_{\text{hom}} = \sum_{i \in \mathcal{V}} \frac{\sum_{j \in \mathcal{N}_i} \hat{p}_{ij} \cdot kl(\hat{\mathbf{Y}}_j||\hat{\mathbf{Y}}_i)}{\sum_{j \in \mathcal{N}_i} \hat{p}_{ij}}$, where $\mathcal{N}_i$ denotes the set of first-order neighbors of node $v_i$ within the estimated latent structure $\hat{\mathbf{A}}$.

**Loss term $\mathbb{E}_{Z_Y \sim q_{\phi_3}}[kl(q_{\phi_2}(\epsilon|X, A, Z_Y)||p(\epsilon))]$.** We decompose $kl(q_{\phi_2}(\epsilon|X, A, Z_Y)||p(\epsilon))$ into two terms: $kl(q_{\phi_{21}}(\epsilon_X|X, Z_Y)||p(\epsilon_X))$ and $kl(q_{\phi_{22}}(\epsilon_A|X, A)||p(\epsilon_A))$. Moreover, the expec-

tation can be removed since we empirically model $q_{\phi_2}(\epsilon|X, A, Z_Y)$ using a deterministic encoder $\text{GCN}_{\phi_3}(\mathbf{X}, \hat{\mathbf{A}})$. We assume that $p(\epsilon_X)$ follows the standard multivariate normal distribution, which means that a closed form solution of $kl(q_{\phi_{21}}(\epsilon_X)||p(\epsilon_X))$ can be obtained as $\mathcal{L}_{\text{p}} = -\frac{1}{2}\sum_{j=1}^{d_2}(1 + \log\sigma_j^2 - \mu_j^2 - \sigma_j^2)$, where $d_2$ is the dimension of a $\epsilon_X$ (Kingma & Welling, 2013). We have prior knowledge about $\epsilon_A$, i.e., $p(\epsilon_A)$, that the loss-based criteria $\hat{p}_{ij}^{el}$ can introduce uncertainty in identifying clean edges as it relies on a single training point's value. In other words, $q_{\phi_{22}}(\epsilon_A|X, A)$ follows an unknown distribution with high variance, while our prior knowledge is that $p(\epsilon_A)$ follows the same distribution with low variance. To reduce uncertainty, i.e., reducing $kl(q_{\phi_{22}}(\epsilon_A|X, A)||p(\epsilon_A))$, we adopt an exponential moving average (EMA) technique: $\hat{p}_{ij}^{el} \leftarrow \xi\hat{p}_{ij}^{el} + (1 - \xi)\hat{p}_{ij}^c$, where $\hat{p}_{ij}^c$ indicates the value of $\hat{p}_{ij}$ at the current training point, and $\xi$ indicates the decaying hyperparameter fixed to 0.9.

**Loss term** $-\mathbb{E}_{Z_A \sim q_{\phi_1}}\mathbb{E}_{\epsilon \sim q_{\phi_2}}[\log(p_{\theta_1}(A|X, \epsilon, Z_A))]$. This term is implemented as an edge reconstruction loss forcing the estimated latent structure $\hat{\mathbf{A}}$ to assign greater weights to clean edges and reduce the influence of noisy edges under the guidance of the positive edges $\mathcal{E}$, which is defined as $\mathcal{L}_{\text{rec-edge}} = \frac{N}{|\mathcal{E}|+|\mathcal{E}^-|}\left(\sum_{(i,j)\in\mathcal{E}}(w_{ij} - \tau_{ij})^2 + \sum_{(i,j)\in\mathcal{E}^-}(\hat{p}_{ij} - 0)^2\right)$, where $\mathcal{E}^-$ denotes randomly sampled negative edges. However, the observed graph structure $A$ contains noisy edges incurred by $X$ and $\epsilon$, which introduce inaccurate supervision. Therefore, we employ regularizations on both the predictions (i.e., $w_{ij}$) and labels (i.e., $\tau_{ij}$) to obtain a robust $\hat{\mathbf{A}}$. The regularized prediction $w_{ij}$ is defined as: $w_{ij} = \theta_1\hat{p}_{ij} + (1 - \theta_1)s(\mathbf{X}_i, \mathbf{X}_j)$. Note that the feature similarity $s(\mathbf{X}_i, \mathbf{X}_j)$ is considered in the prediction of positive edges. The main idea is to penalize $\hat{p}_{ij}$ when $s(\mathbf{X}_i, \mathbf{X}_j)$ is high, as the edge between $v_i$ and $v_j$ is potentially noisy due to the influence of noisy $X$. To create regularized labels, we convert $\hat{p}_{ij}^{el}$ into $\tau_{ij}$ with a minimum value of 0.9 and a maximum value of 1. In other words, when an edge is regarded as noisy (i.e., with a low $\hat{p}_{ij}^{el}$), its label is close to 0.9, while an edge considered clean (i.e., with a high $\hat{p}_{ij}^{el}$) has a label close to 1. This approach achieves a similar effect to label smoothing, enhancing robustness in the presence of noisy supervision. Note that $\theta_1$ is a hyperparameter, and the value 0.9 is selected in the label regularization following Szegedy et al. (2016).

**Loss term** $-\mathbb{E}_{\epsilon \sim q_{\phi_2}}\mathbb{E}_{Z_Y \sim q_{\phi_3}}[\log(p_{\theta_2}(X|\epsilon, Z_Y))]$. This term is implemented as a feature reconstruction loss $\mathcal{L}_{\text{rec-feat}}$, where the decoder $p_{\theta_2}$ is composed of an MLP that takes $CONCAT(\epsilon_X, Z_Y)$ as an input and outputs reconstructed node features. Note that the reparametrization trick (Kingma & Welling, 2013) is used for sampling $\epsilon_X$ that follows the standard normal distribution. To minimize $\mathcal{L}_{\text{rec-feat}}$, the decoder needs to rely on the information contained in $Z_Y$, which essentially encourages the value of $Z_Y$ to be meaningful for the prediction process, i.e., generating $X$.

**Loss term** $-\mathbb{E}_{Z_Y \sim q_{\phi_3}}[\log(p_{\theta_3}(Y|X, A, Z_Y))]$. The probability $p(Y|X, A, Z_Y)$ represents the transition relationship from the latent clean label $\bar{Z}_Y$ to the noisy label $Y$ of an instance, i.e., how the label noise was generated (Yao et al., 2021). For this reason, maximizing $p_{\theta_3}(Y|X, A, Z_Y)$ (or equivalently minimizing the loss term) would let us discover the latent true label $Z_Y$ from which the noisy label $Y$ is generated given an instance, i.e., $X$ and $A$. We implement the loss term as a node classification loss $\mathcal{L}_{\text{cls-dec}}$, i.e., the cross entropy loss, where the decoder $p_{\theta_3}$ is composed of a GCN classifier that takes $A$ and $CONCAT(X, Z_Y)$ as inputs and outputs the prediction of $Y$, i.e., $\hat{\mathbf{Y}}_{\text{dec}} = \text{GCN}_{\theta_3}(\mathbf{X}, \mathbf{A}, \hat{\mathbf{Y}}) \in \mathbb{R}^{N \times C}$.

In summary, the overall learning objective can be written as follows and PRINGLE is trained to minimize the $\mathcal{L}_{\text{final}}$:

$$\mathcal{L}_{\text{final}} = \mathcal{L}_{\text{cls-enc}} + \lambda_1\mathcal{L}_{\text{rec-edge}} + \lambda_2\mathcal{L}_{\text{hom}} + \lambda_3(\mathcal{L}_{\text{rec-feat}} + \mathcal{L}_{\text{cls-dec}} + \mathcal{L}_{\text{p}}), \quad (4)$$

where $\lambda_1$, $\lambda_2$, and $\lambda_3$ are the balancing coefficients.

## 5 EXPERIMENTS

**Datasets.** We evaluate PRINGLE and baselines on **four commonly used benchmark datasets** (i.e., Cora, Citeseer, Photo, and Computers) and **two newly introduced datasets** (i.e., Auto and Garden). Auto and Garden are proposed in this work based on Amazon review data (He & McAuley, 2016; McAuley et al., 2015) to mimic FDGN on e-commerce systems (Refer to Appendix D.2.2 for details). The details of the datasets are given in Appendix D.1.

**Experimental Details.** We evaluated PRINGLE in both node classification and link prediction tasks, comparing it with noise-robust graph learning and generative graph learning methods. For a

Table 1: Node classification performance under synthetic feature-dependent graph-noise (FDGN).

| Dataset | Setting | WSGNN | AirGNN | ProGNN | RSGNN | STABLE | EvenNet | NRGNN | RTGNN | PRINGLE |
|---------|---------|-------|--------|--------|-------|--------|---------|-------|-------|---------|
| Cora | Clean | **86.2±0.1** | 85.0±0.2 | 85.3±0.4 | **86.2±0.5** | 86.1±0.2 | **86.2±0.0** | **86.2±0.2** | 86.1±0.2 | **86.2±0.7** |
| | FDGN-10% | 80.7±0.3 | 79.7±0.5 | 79.6±0.7 | 81.9±0.3 | 82.2±0.7 | 80.7±0.7 | 81.0±0.5 | 81.6±0.5 | **82.9±0.6** |
| | FDGN-30% | 70.0±0.6 | 71.5±0.8 | 74.5±0.1 | 71.9±0.5 | 74.3±0.3 | 65.2±1.7 | 73.5±0.8 | 72.1±0.6 | **78.2±0.3** |
| | FDGN-50% | 55.9±1.1 | 56.2±0.8 | 66.4±0.4 | 58.1±0.2 | 62.8±2.4 | 47.1±1.8 | 61.9±1.4 | 60.8±0.4 | **69.7±0.6** |
| Citeseer | Clean | 76.6±0.6 | 71.5±0.2 | 72.6±0.5 | 75.8±0.4 | 74.6±0.6 | 76.4±0.5 | 75.0±1.3 | 76.1±0.4 | **77.3±0.6** |
| | FDGN-10% | 72.8±0.8 | 66.2±0.7 | 67.5±0.6 | 73.3±0.5 | 71.5±0.3 | 71.1±0.4 | 71.9±0.3 | 73.2±0.2 | **74.3±0.9** |
| | FDGN-30% | 63.3±0.7 | 58.0±0.4 | 61.0±0.2 | 63.9±0.5 | 62.5±1.4 | 61.2±0.6 | 62.5±0.7 | 63.5±2.1 | **65.6±0.6** |
| | FDGN-50% | 53.4±0.6 | 50.0±0.6 | 53.3±0.2 | 55.3±0.4 | 54.7±1.7 | 47.2±1.1 | 52.6±0.9 | 54.2±1.8 | **59.0±1.8** |
| Photo | Clean | 92.9±0.3 | 93.5±0.1 | 90.1±0.2 | 93.6±0.8 | 93.4±0.1 | 94.5±0.4 | 90.3±1.7 | 91.3±0.6 | **94.8±0.3** |
| | FDGN-10% | 83.9±1.8 | 87.3±0.9 | 84.3±0.1 | 89.4±2.4 | 92.2±0.1 | 92.6±0.0 | 84.3±1.3 | 88.9±0.3 | **93.2±0.2** |
| | FDGN-30% | 51.9±6.8 | 67.8±4.3 | 74.7±0.2 | 82.1±1.1 | 88.0±1.0 | 89.6±0.2 | 69.0±2.2 | 86.4±0.5 | **90.5±0.4** |
| | FDGN-50% | 31.9±5.6 | 57.8±0.7 | 48.9±0.5 | 75.6±2.6 | 80.2±1.8 | 84.6±0.4 | 57.5±1.8 | 79.2±0.3 | **87.6±0.2** |
| Comp | Clean | 83.1±3.1 | 83.4±1.2 | 83.9±0.8 | 91.1±0.1 | 90.2±0.2 | 90.1±0.2 | 87.5±1.0 | 87.3±1.0 | **92.2±0.0** |
| | FDGN-10% | 75.0±1.2 | 76.8±1.8 | 72.0±0.2 | 86.0±1.9 | 85.9±0.5 | 87.6±0.7 | 85.7±0.9 | 81.7±0.2 | **89.8±0.2** |
| | FDGN-30% | 48.5±5.8 | 59.2±0.9 | 66.9±0.8 | 81.5±1.7 | 80.4±1.0 | 84.8±0.5 | 74.8±3.5 | 73.9±0.2 | **86.9±0.3** |
| | FDGN-50% | 39.6±4.0 | 44.1±1.4 | 43.3±0.3 | 73.9±2.3 | 68.8±1.3 | 77.5±1.9 | 65.3±3.2 | 68.5±0.3 | **82.2±0.4** |

thorough evaluation, we create synthetic and real-world FDGN settings, with details in Appendix D.2. We also account for independent structure/feature/label noise that are also prevalent in real-world applications, following Li et al. (2022a); Liu et al. (2021); Qian et al. (2023). Due to the space limit, we provide additional analyses on the model robustness under independent structure/feature/label noise in Appendix E.1. Further details about the baselines, evaluation protocol, and implementation details can be found in Appendix D.3, D.4, and D.5, respectively.

## 5.1 QUANTITATIVE RESULTS

### 5.1.1 UNDER SYNTHETIC FEATURE-DEPENDENT GRAPH-NOISE

We first evaluate PRINGLE under synthetic FDGN settings. Appendix D.2.1 provides a description of the synthetic data generation algorithm. Table 1 shows that PRINGLE consistently outperforms all baselines in FDGN scenarios, especially when noise levels are high. This superiority is attributed to the fact that PRINGLE captures the causal relationships involved in the DGP of FDGN, while the baselines overlook such relationships, leading to their model designs assuming the completeness of at least one data source. Additionally, PRINGLE performs well even in clean graph settings. We attribute this to the accurate inference of $\epsilon_A$, which is utilized as the label regularization in calculating $\mathcal{L}_{\text{rec-edge}}$. Specifically, in Fig 12(a) in Appendix E.4, we observe that the $\hat{p}_{ij}^{el}$ values estimated from the clean graph tend to be close to 1, while those from the graph with FDGN are considerably smaller. Recall that the high value of $\hat{p}_{ij}^{el}$ indicates the model regards the edge $(i, j)$ as a clean edge. This suggests that PRINGLE has the capability to adapt its model learning to the level of noise present in the input graph, resulting in superior performance on both clean and noisy graphs.

Table 2: Node classification performance under real-world FDGN.

| Methods | Auto | | Garden | |
|---------|------|------|--------|------|
| | Clean | + FDGN | Clean | + FDGN |
| WSGNN | 71.8±4.3 | 57.7±1.3 | 87.4±0.2 | 77.6±0.8 |
| AirGNN | 69.5±0.8 | 53.9±0.1 | 78.3±1.5 | 66.1±1.7 |
| ProGNN | 63.2±0.2 | 48.6±0.3 | 78.7±0.1 | 73.0±0.4 |
| RSGNN | 69.5±0.4 | 56.8±0.9 | 83.3±1.2 | 76.2±0.5 |
| STABLE | 71.6±0.9 | 57.5±0.2 | 84.2±0.4 | 77.2±3.3 |
| EvenNet | 73.4±0.5 | 57.1±2.1 | 85.7±0.5 | 75.6±2.4 |
| NRGNN | 74.3±0.8 | 55.8±1.0 | 87.7±0.4 | 76.1±0.2 |
| RTGNN | 75.1±0.3 | 59.6±0.8 | 85.5±0.2 | 76.0±0.6 |
| PRINGLE | **79.3±0.2** | **61.4±0.4** | **88.7±0.3** | **80.2±0.8** |

Table 3: Link prediction performance under real-world FDGN.

| Methods | Auto | | Garden | |
|---------|------|------|--------|------|
| | Clean | + FDGN | Clean | + FDGN |
| WSGNN | 81.8±0.1 | 69.1±0.6 | 84.7±0.2 | 84.6±0.7 |
| AirGNN | 60.2±0.2 | 57.9±0.4 | 62.0±0.1 | 58.2±0.5 |
| ProGNN | 74.8±0.3 | 56.7±0.5 | 83.5±0.6 | 83.3±0.5 |
| RSGNN | 87.2±0.8 | 65.0±0.2 | 91.2±0.4 | 91.2±0.5 |
| STABLE | 78.6±0.1 | 57.3±0.1 | 85.2±0.2 | 85.0±0.1 |
| EvenNet | 86.8±0.1 | 70.5±0.2 | 89.2±0.3 | 90.0±0.7 |
| NRGNN | 76.6±1.3 | 47.5±1.7 | 87.0±0.9 | 58.6±4.5 |
| RTGNN | 84.4±0.1 | 72.2±0.2 | 90.4±0.3 | 90.4±0.2 |
| PRINGLE | **88.2±0.3** | **73.6±0.6** | **92.6±0.2** | **92.4±0.4** |

### 5.1.2 UNDER REAL-WORLD FEATURE-DEPENDENT GRAPH-NOISE

To investigate the robustness of PRINGLE under real-world noise scenarios, we newly design two new benchmark graph datasets, i.e., Auto and Garden, where the node label is the product category, the node feature is bag-of-words representation of product reviews, and the edges indicate the co-purchase relationship between two products by the same user. To the best of our knowledge, this is the first work proposing new datasets for evaluating the noise-robust graph learning under realistic noise scenarios that are plausible in a real-world e-commerce system containing fraudsters. Appendix D.2.2 provides a comprehensive description of the data generation algorithm. In Table 2 and 3, we observe that PRINGLE outperforms the baselines under FDGN caused by malicious actions of fraudsters on both the node classification and link prediction tasks. This indicates that PRINGLE works well not

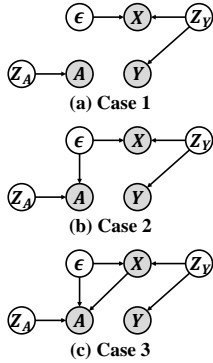

(a) Case 1

(b) Case 2

(c) Case 3

Figure 4: Graphical models of DGPs derived from FDGN.

Table 4: Ablation studies regarding various DGPs in Fig 4. In Case 3, the causal relationship $Y \leftarrow (X, A)$ is removed from the DGP of FDGN (See Fig 2(b)). In Case 2, $A \leftarrow X$ is additionally removed. In Case 1, $A \leftarrow \epsilon$ is additionally removed, which is equivalent to the DGP of CGN (See Fig 2(a)). Cora and Citeseer datasets are used to evaluate the node classification performance.

| Dataset | Setting | (a) Case 1 | (b) Case 2 | (c) Case 3 | PRINGLE |
|---------|---------|-----------|-----------|-----------|---------|
| Cora | Clean | 84.6±0.4 | 84.8±0.4 | **86.2±0.2** | **86.2±0.7** |
| | FDGN-10% | 77.4±0.3 | 77.3±0.3 | **83.2±0.3** | 82.9±0.6 |
| | FDGN-30% | 68.3±0.4 | 68.5±0.2 | 77.3±0.4 | **78.2±0.3** |
| | FDGN-50% | 55.2±0.2 | 56.1±0.3 | 68.7±0.3 | **69.7±0.6** |
| Citeseer | Clean | 76.7±0.9 | 76.8±0.8 | 76.5±0.9 | **77.3±0.6** |
| | FDGN-10% | 69.5±0.3 | 69.5±0.4 | 73.2±0.1 | **74.3±0.9** |
| | FDGN-30% | 57.2±1.1 | 57.7±0.5 | 65.5±0.7 | **65.6±0.6** |
| | FDGN-50% | 49.2±0.5 | 48.7±0.2 | 57.6±2.5 | **59.0±1.8** |

only under artificially generated noise, but also under noise scenarios that are plausible in real-world applications.

## 5.2 ABLATION STUDIES

To emphasize the importance of directly capturing the causal relationships among variables in the DGP of FDGN, i.e., $Y \leftarrow (X, A)$, $A \leftarrow X$, and $A \leftarrow \epsilon$, we remove them one by one from the graphical model of FDGN (See Fig 2(b)), and then design deep generative models based on the DGPs in a similar manner to PRINGLE. The graphical models of the derived DGPs are illustrated in Fig 4. In Table 4, we observe that as more causal relationships are removed from the DGP of FDGN, the node classification performance decreases. Below, we offer explanations for this observation from the perspective of model derivation.

**1)** By removing $Y \leftarrow (X, A)$, i.e., Fig 4(c), the loss term $-\mathbb{E}_{Z_Y \sim q_{\phi_3}} [\log(p_{\theta_3}(Y|X, A, Z_Y))]$ can be simplified to $-\mathbb{E}_{Z_Y \sim q_{\phi_3}} [\log(p_{\theta_3}(Y|Z_Y))]$. This simplification hinders the accurate modeling of the label transition relationship from $Z_Y$ to the noisy label $Y$, resulting in a degradation of model performance under FDGN. **2)** Additionally, when eliminating $A \leftarrow X$ (i.e., Fig 4(b)), the inference of $Z_A$ and $Z_Y$ is simplified as follows: $q_{\phi_1}(Z_A|X, A)$ to $q_{\phi_1}(Z_A|A)$ and $q_{\phi_3}(Z_Y|X, A)$ to $q_{\phi_3}(Z_Y|X)$. Furthermore, the loss term $-\mathbb{E}_{Z_A \sim q_{\phi_1}} \mathbb{E}_{\epsilon \sim q_{\phi_2}} [\log(p_{\theta_1}(A|X, \epsilon, Z_A))]$ is also simplified to $-\mathbb{E}_{Z_A \sim q_{\phi_1}} \mathbb{E}_{\epsilon \sim q_{\phi_2}} [\log(p_{\theta_1}(A|\epsilon, Z_A))]$. These simplifications significantly hinder the accurate inference of $Z_A$ and $Z_Y$, resulting in a notable performance degradation. **3)** Furthermore, by eliminating $A \leftarrow \epsilon$, i.e., Fig 4(a), the loss term $-\mathbb{E}_{Z_A \sim q_{\phi_1}} \mathbb{E}_{\epsilon \sim q_{\phi_2}} [\log(p_{\theta_1}(A|\epsilon, Z_A))]$ is simplified to $-\mathbb{E}_{Z_A \sim q_{\phi_1}} \mathbb{E}_{\epsilon \sim q_{\phi_2}} [\log(p_{\theta_1}(A|Z_A))]$. This simplification hinders the robustness of the inferred $Z_A$, since the simplified loss excludes label regularization from the model training process, ultimately resulting in performance degradation.

## 6 CONCLUSION AND FUTURE WORK

In this paper, we discover practical limitations of conventional graph noise in terms of node features, i.e., the noise in node features is independent of the graph structure or node label. To mitigate limitations of the conventional graph noise assumption, we introduce a more realistic graph noise scenario called feature-dependent graph-noise (FDGN), and present a deep generative model that effectively captures the causal relationships among variables in the DGP of FDGN. Our proposed method, PRINGLE, consistently outperforms baselines in both node classification and link prediction tasks. We evaluate PRINGLE on commonly used benchmark datasets and newly introduced real-world graph datasets that simulate FDGN in e-commerce systmes, which is expected to foster practical research in noise-robust graph learning. For future work, the practicality of FDGN can be further enhanced by additionally considering the causal relationship $X \leftarrow A$, which indicates that the graph structure noise inevitably entails the node feature noise, that may indeed manifest in some real-world scenarios. Since it can cover a broader range of noise scenarios that occur in real-world applications than FDGN, we expect directly modeling it has the potential to enhance practical applicability. We provide a detailed discussion on this topic in Appendix F.

## ETHICS STATEMENT

Regarding the adherence of ICLR Code of Ethics, to the best of our knowledge, there are no ethical issues with this paper. All datasets used for experiments are publicly available.

## REPRODUCIBILITY STATEMENT

To ensure clarity and reproducibility, in Sec 4.3, we offer comprehensive explanations of our proposed method (PRINGLE). Implementation details for both baseline methods and our method can be found in Appendix D.3 and D.5. Furthermore, as we introduce novel settings to address the more realistic graph noise scenario, FDGN, we provide a comprehensive discussion of the problem formulation in Sec 3.1 and elaborate the experimental setup construction procedure in Appendix D.2 Our code is available at https://anonymous.4open.science/r/FDGN-PRINGLE-4B31/

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

# Supplementary Material

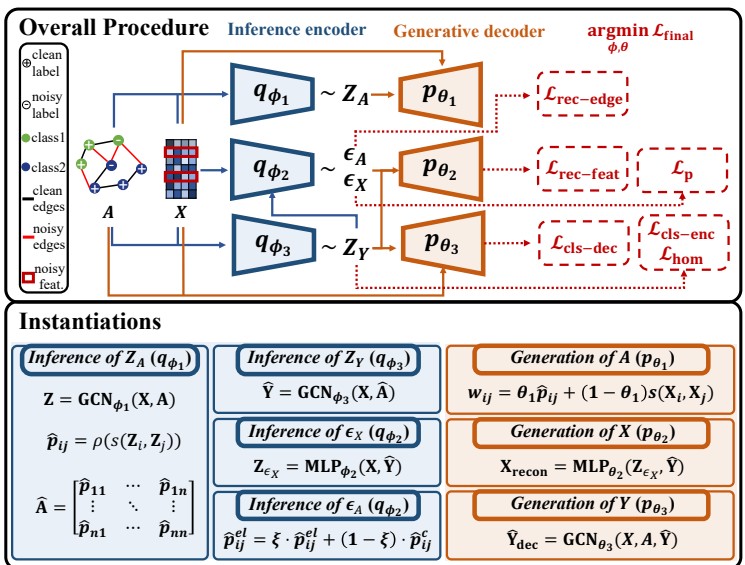

Figure 5: Overall architecture of PRINGLE.

## A  OVERALL ARCHITECTURE AND ALGORITHM

Fig. 5 shows the overall architecture of PRINGLE, and the detailed algorithm is provided in Algorithm. 1.

## B  DERIVATION DETAILS OF EVIDENCE LOWER BOUND (ELBO)

In this section, we derive the Evidence Lower BOund (ELBO) for the observed data log-likelihood $P(X, A, Y)$. First, we factorize the joint distribution $P(X, A, Y, \epsilon, Z_A, Z_Y)$ based on the graphical model in Fig. 2(b) in the main paper:

$$P(X, A, Y, \epsilon, Z_A, Z_Y) = P(\epsilon)P(Z_A)P(Z_Y)P(X|\epsilon, Z_Y)P(A|\epsilon, X, Z_A)P(Y|X, A, Z_Y). \quad (5)$$

Thus, the conditional distribution $P_\theta(X, A, Y|\epsilon, Z_A, Z_Y)$ can be represented as follows:

$$P_\theta(X, A, Y|\epsilon, Z_A, Z_Y) = P_{\theta_1}(X|\epsilon, Z_Y)P_{\theta_2}(A|\epsilon, X, Z_A)P_{\theta_3}(Y|X, A, Z_Y). \quad (6)$$

Recall that the conditional distribution $q_\phi(\epsilon, Z_A, Z_Y|X, A, Y)$ is factorized as in Eqn. 2 in the main paper:

$$q_\phi(\epsilon, Z_A, Z_Y|X, A, Y) = q_{\phi_1}(Z_A|X, A)q_{\phi_2}(\epsilon|X, A, Z_Y)q_{\phi_3}(Z_Y|X, A). \quad (7)$$

Now, we derive the ELBO for the observed data log-likelihood $P(X, A, Y)$:

$$\log p_\theta(X,A,Y) = \log \int_\epsilon \int_{Z_A} \int_{Z_Y} p_\theta(X,A,Y,\epsilon,Z_A,Z_Y) d\epsilon dZ_A dZ_Y$$

$$= \log \int_\epsilon \int_{Z_A} \int_{Z_Y} p_\theta(X,A,Y,\epsilon,Z_A,Z_Y) \frac{q_\phi(\epsilon,Z_A,Z_Y|X,A,Y)}{q_\phi(\epsilon,Z_A,Z_Y|X,A,Y)} d\epsilon dZ_A dZ_Y$$

$$= \log \mathbb{E}_{(\epsilon,Z_A,Z_Y)\sim q_\phi(\epsilon,Z_A,Z_Y|X,A,Y)} \left[ \frac{p_\theta(X,A,Y,\epsilon,Z_A,Z_Y)}{q_\phi(\epsilon,Z_A,Z_Y|X,A,Y)} \right]$$

$$\geq \mathbb{E}_{(\epsilon,Z_A,Z_Y)\sim q_\phi(\epsilon,Z_A,Z_Y|X,A,Y)} \left[ \log \frac{p_\theta(X,A,Y,\epsilon,Z_A,Z_Y)}{q_\phi(\epsilon,Z_A,Z_Y|X,A,Y)} \right] := \text{ELBO}$$

$$= \mathbb{E}_{(\epsilon,Z_A,Z_Y)\sim q_\phi(\epsilon,Z_A,Z_Y|X,A,Y)} \left[ \log \frac{p(\epsilon)p(Z_A)p(Z_Y)p_{\theta_1}(A|X,\epsilon,Z_A)p_{\theta_2}(X|\epsilon,Z_Y)p_{\theta_3}(Y|X,A,Z_Y)}{q_\phi(\epsilon,Z_A,Z_Y|X,A,Y)} \right]$$

$$= \mathbb{E}_{(\epsilon,Z_A,Z_Y)\sim q_\phi(\epsilon,Z_A,Z_Y|X,A,Y)} \left[ \log(p_{\theta_1}(A|X,\epsilon,Z_A)) + \log(p_{\theta_2}(X|\epsilon,Z_Y)) + \log(p_{\theta_3}(Y|X,A,Z_Y)) \right]$$

$$+ \mathbb{E}_{(\epsilon,Z_A,Z_Y)\sim q_\phi(\epsilon,Z_A,Z_Y|X,Y,A)} \left[ \log \frac{p(\epsilon)p(Z_A)p(Z_Y)}{q_{\phi_1}(Z_A|X,A)q_{\phi_2}(\epsilon|X,A,Z_Y)q_{\phi_3}(Z_Y|X,A)} \right] \quad (8)$$

The last equation of Eq. 8 can be more simplified as follows:

$$\mathbb{E}_{(\epsilon,Z_A,Z_Y)\sim q_\phi(\epsilon,Z_A,Z_Y|X,A,Y)} \left[ \log(p_{\theta_1}(A|X,\epsilon,Z_A)) \right]$$

$$= \mathbb{E}_{Z_A\sim q_{\phi_1}(Z_A|X,A)} \mathbb{E}_{\epsilon\sim q_{\phi_2}(\epsilon|X,A,Z_Y)} \mathbb{E}_{Z_Y\sim q_{\phi_3}(Z_Y|X,A)} \left[ \log(p_{\theta_1}(A|X,\epsilon,Z_A)) \right]$$

$$= \mathbb{E}_{Z_A\sim q_{\phi_1}(Z_A|X,A)} \mathbb{E}_{\epsilon\sim q_{\phi_2}(\epsilon|X,A,Z_Y)} \left[ \log(p_{\theta_1}(A|X,\epsilon,Z_A)) \right], \quad (9)$$

and

$$\mathbb{E}_{(\epsilon,Z_A,Z_Y)\sim q_\phi(\epsilon,Z_A,Z_Y|X,A,Y)} \left[ \log(p_{\theta_2}(X|\epsilon,Z_Y)) \right]$$

$$= \mathbb{E}_{Z_A\sim q_{\phi_1}(Z_A|X,A)} \mathbb{E}_{\epsilon\sim q_{\phi_2}(\epsilon|X,A,Z_Y)} \mathbb{E}_{Z_Y\sim q_{\phi_3}(Z_Y|X,A)} \left[ \log(p_{\theta_2}(X|\epsilon,Z_Y)) \right]$$

$$= \mathbb{E}_{\epsilon\sim q_{\phi_2}(\epsilon|X,A,Z_Y)} \mathbb{E}_{Z_Y\sim q_{\phi_3}(Z_Y|X,A)} \left[ \log(p_{\theta_2}(X|\epsilon,Z_Y)) \right], \quad (10)$$

and

$$\mathbb{E}_{(\epsilon,Z_A,Z_Y)\sim q_\phi(\epsilon,Z_A,Z_Y|X,A,Y)} \left[ \log(p_{\theta_3}(Y|X,A,Z_Y)) \right]$$

$$= \mathbb{E}_{Z_A\sim q_{\phi_1}(Z_A|X,A)} \mathbb{E}_{\epsilon\sim q_{\phi_2}(\epsilon|X,A,Z_Y)} \mathbb{E}_{Z_Y\sim q_{\phi_3}(Z_Y|X,A)} \left[ \log(p_{\theta_3}(Y|X,A,Z_Y)) \right]$$

$$= \mathbb{E}_{Z_Y\sim q_{\phi_3}(Z_Y|X,A)} \left[ \log(p_{\theta_3}(Y|X,A,Z_Y)) \right]. \quad (11)$$

In a similar way, the last term can be also simplified:

$$\mathbb{E}_{(\epsilon,Z_A,Z_Y)\sim q_\phi(\epsilon,Z_A,Z_Y|X,Y,A)} \left[ \log \frac{p(\epsilon)p(Z_A)p(Z_Y)}{q_{\phi_1}(Z_A|X,A)q_{\phi_2}(\epsilon|X,A,Z_Y)q_{\phi_3}(Z_Y|X,A)} \right]$$

$$= \mathbb{E}_{Z_A\sim q_{\phi_1}(Z_A)} \mathbb{E}_{\epsilon\sim q_{\phi_2}(\epsilon|Z_Y)} \mathbb{E}_{Z_Y\sim q_{\phi_3}(Z_Y)} \left[ \log \frac{p(Z_A)p(\epsilon)p(Z_Y)}{q_{\phi_1}(Z_A)q_{\phi_2}(\epsilon|Z_Y)q_{\phi_3}(Z_Y)} \right]$$

$$= \mathbb{E}_{Z_A\sim q_{\phi_1}(Z_A)} \mathbb{E}_{\epsilon\sim q_{\phi_2}(\epsilon|Z_Y)} \mathbb{E}_{Z_Y\sim q_{\phi_3}(Z_Y)} \left[ \log \frac{p(Z_A)}{q_{\phi_1}(Z_A)} + \log \frac{p(\epsilon)}{q_{\phi_2}(\epsilon|Z_Y)} + \log \frac{p(Z_Y)}{q_{\phi_3}(Z_Y)} \right]$$

$$= \mathbb{E}_{Z_A\sim q_{\phi_1}(Z_A)} \left[ \log \frac{p(Z_A)}{q_{\phi_1}(Z_A)} \right] + \mathbb{E}_{\epsilon\sim q_{\phi_2}(\epsilon|Z_Y)} \mathbb{E}_{Z_Y\sim q_{\phi_3}(Z_Y)} \left[ \log \frac{p(\epsilon)}{q_{\phi_2}(\epsilon|Z_Y)} \right]$$

$$+ \mathbb{E}_{Z_Y\sim q_{\phi_3}(Z_Y)} \left[ \log \frac{p(Z_Y)}{q_{\phi_3}(Z_Y)} \right]$$

$$= -kl(q_{\phi_1}(Z_A)||p(Z_A)) - \mathbb{E}_{Z_Y\sim q_{\phi_3}(Z_Y)} \left[ kl(q_{\phi_2}(\epsilon|Z_Y)||p(\epsilon)) \right] - kl(q_{\phi_3}(Z_Y)||p(Z_Y)) \quad (12)$$

where we abuse the notation $q_{\phi_1}(Z_A|X, A)$, $q_{\phi_2}(\epsilon|X, A, Z_Y)$, and $q_{\phi_3}(Z_Y|X, A)$ as $q_{\phi_1}(Z_A)$, $q_{\phi_2}(\epsilon|Z_Y)$, and $q_{\phi_3}(Z_Y)$, respectively. We combine Eqn. 9, 10, 11, and 12 to get the negative ELBO, i.e., $\mathcal{L}_{\text{ELBO}}$:

$$
\begin{aligned}
\mathcal{L}_{\text{ELBO}} = \\
- \mathbb{E}_{Z_A \sim q_{\phi_1}(Z_A|X,A)} \mathbb{E}_{\epsilon \sim q_{\phi_2}(\epsilon|X,A,Z_Y)} \left[\log(p_{\theta_1}(A|X, \epsilon, Z_A))\right] + kl(q_{\phi_1}(Z_A|X,A)||p(Z_A)) \\
- \mathbb{E}_{\epsilon \sim q_{\phi_2}(\epsilon|X,A,Z_Y)} \mathbb{E}_{Z_Y \sim q_{\phi_3}(Z_Y|X,A)} \left[\log(p_{\theta_2}(X|\epsilon, Z_Y))\right] + \mathbb{E}_{Z_Y \sim q_{\phi_3}(Z_Y|X,A)} \left[kl(q_{\phi_2}(\epsilon|X,A,Z_Y)||p(\epsilon))\right] \\
- \mathbb{E}_{Z_Y \sim q_{\phi_3}(Z_Y|X,A)} \left[\log(p_{\theta_3}(Y|X, A, Z_Y))\right] + kl(q_{\phi_3}(Z_Y|X,A)||p(Z_Y))
\end{aligned}
\tag{13}
$$

# C   DETAILS OF MODEL INSTANTIATIONS

**Loss term** $kl(q_{\phi_1}(Z_A|X, A)||p(Z_A))$. To minimize this term, we encourage $Z_A$ to align with our prior knowledge, i.e., $p(Z_A)$, that the latent graph structure predominantly consists of assortative edges that have the potential to enhance feature propagation within GNNs (Zhao et al., 2023). In recent studies (Choi et al., 2022; Dai et al., 2022), the $\gamma$-hop subgraph similarity has served as a potent metric for identifying assortative edges. Hence, we model $q_{\phi_1}(Z_A|X, A)$ to minimize the KL divergence between the latent graph structure $\hat{\mathbf{A}}$ and prior graph structure $\mathbf{A}^{\text{p}} = \{a_{ij}^{\text{p}}\}_{N \times N}$, where $a_{ij}^{\text{p}}$ is sampled from a Bernoulli distribution with a probability as given by the $\gamma$-hop subgraph similarity. This leads the estimated probability $\hat{p}_{ij}$ between two nodes to increase if they exhibit a high subgraph similarity. However, computing $\hat{p}_{ij}$ in every epoch is impractical for large graphs, i.e., $O(N^2)$. To mitigate the issue, we pre-define a candidate graph that consists of the observed edge set $\mathcal{E}$ and a $k$-NN graph based on the $\gamma$-hop subgraph similarity. We denote the set of edges in the $k$-NN graphs as $\mathcal{E}_k^{\gamma}$. Then, we compute the $\hat{p}_{ij}$ values of the edges in a candidate graph, i.e., $\mathcal{E}_k^{\gamma} \cup \mathcal{E}$, instead of all edges in $\{(i, j)|i \in \mathcal{V}, j \in \mathcal{V}\}$, to estimate the latent graph structure denoted as $\hat{\mathbf{A}}$. It is important to highlight that obtaining $\mathcal{E}_k^{\gamma}$ is carried out offline before model training, thus incurring no additional computational overhead during training. This implementation technique achieves a similar effect as minimizing $kl(q_{\phi_1}(Z_A|X, A)||p(Z_A))$ while significantly addressing computational complexity from $O(N^2)$ to $O(|\mathcal{E}_k^{\gamma} \cup \mathcal{E}|)$, where $N^2 \gg |\mathcal{E}_k^{\gamma} \cup \mathcal{E}|$.

# D   DETAILS ON EXPERIMENTAL SETTINGS

## D.1   DATASETS

We evaluate PRINGLE and baselines on **four existing datasets** (i.e., Cora (Yang et al., 2016), Citeseer (Yang et al., 2016), Amazon Photo, and Amazon Computers (Shchur et al., 2018)) and **two newly introduced datasets** (i.e., Amazon Auto and Amazon Garden) that are proposed in this work based on Amazon review data (He & McAuley, 2016; McAuley et al., 2015) to mimic FDGN caused by malicious fraudsters on e-commerce systems (Refer to Appendix D.2.2 for details). The statistics of the datasets are given in Table 5. These six datasets can be found in these URLs:

- **Cora**: https://github.com/ChandlerBang/Pro-GNN/
- **Citeseer**: https://github.com/ChandlerBang/Pro-GNN/
- **Photo**: https://pytorch-geometric.readthedocs.io/en/latest/
- **Computers**: https://pytorch-geometric.readthedocs.io/en/latest/
- **Auto**: http://jmcauley.ucsd.edu/data/amazon/links.html
- **Garden**: http://jmcauley.ucsd.edu/data/amazon/links.html

## D.2   DETAILS OF GENERATING FDGN

### D.2.1   SYNTHETIC FDGN

For the synthetic FDGN settings, we artificially generate the noise following the data generation process of the proposed FDGN scenario. First, we randomly sample a subset of nodes $\mathcal{V}^{\text{noisy}}$ (i.e.,

Table 5: Statistics for datasets.

| Dataset | # Nodes | # Edges | # Features | # Classes |
|---------|---------|---------|-----------|-----------|
| Cora | 2,485 | 5,069 | 1,433 | 7 |
| Citeseer | 2,110 | 3,668 | 3,703 | 6 |
| Photo | 7,487 | 119,043 | 745 | 8 |
| Computers | 13,381 | 245,778 | 767 | 10 |
| Auto | 8,175 | 13,371 | 300 | 5 |
| Garden | 7,902 | 19,383 | 300 | 5 |

10%, 30%, and 50% of the whole node set $\mathcal{V}$). To inject node feature noise into the sampled nodes, we randomly flip 0/1 value on each dimension of node features $\mathbf{X}_i$ from Bernoulli distribution with probability $p = \frac{1}{F}\sum_{i=1}^{F}\mathbf{X}_i$, which results in the noisy features $\mathbf{X}_i^{\text{noisy}}$. After injecting the feature noise, we generate a feature-dependent structure noise (i.e., $A \leftarrow X$) and feature-dependent label noise (i.e., $Y \leftarrow (X, A)$). For the feature-dependent structure noise, we first calculate the similarity vector for each node $v_i$ as $\{s(\mathbf{X}_i^{\text{noisy}}, \mathbf{X}_j)|v_i \in \mathcal{V}^{\text{noisy}}, v_j \in \mathcal{V}\}$ where $s(\cdot, \cdot)$ is a cosine similarity function, and select the node pairs whose feature similarity is top-$k$ highest values. We add the selected node pairs to the original edge set $\mathcal{E}$, which results in $\mathcal{E}^{\text{noisy}}$. To address feature-dependent label noise, we replace the labels of labeled nodes (i.e., training and validation nodes) with randomly sampled labels from a Multinomial distribution, with parameters determined by the normalized neighborhood class distribution. Finally, for the independent structure noise (i.e., $A \leftarrow \epsilon$), we add the randomly selected non-connected node pairs to the $\mathcal{E}^{\text{noisy}}$. Detailed algorithm is provided in Algorithm 2.

### D.2.2 REAL-WORLD FDGN

We have introduced and released two new graph benchmark datasets, i.e., Auto and Garden, that simulate real-world FDGN scenarios on e-commerce systems. To construct these graphs, we utilized metadata and product review data from two categories, "Automotives" and "Patio, Lawn and Garden," obtained from Amazon product review data sources (He & McAuley, 2016; McAuley et al., 2015). Specifically, we generated a clean product-product graph where node features are represented using a bag-of-words technique applied to product reviews. The edges indicate co-purchase relationships between products that have been purchased by the same user, and the node labels correspond to product categories. We perform both node classification and link prediction tasks, which are equivalent to categorizing products and predicting co-purchase relationships, respectively.

We simulate the behaviors of fraudsters on a real-world e-commerce platform that incurs FDGN. When the fraudsters engage with randomly selected products (i.e., when they write fake product reviews), it would make other users purchase irrelevant products, which introduces a substantial number of malicious co-purchase edges within the graph structure. Additionally, this activity involves the injection of noisy random reviews into the node features. To provide a more detailed description, we designated 100 uers as fraudsters. Furthermore, each of these users was responsible for generating 10 fraudulent reviews in both the Auto and Garden datasets. To generate fake review content, we randomly choose text from existing reviews and duplicate it for the targeted products. This approach guarantees that the fake reviews closely mimic the writing style and content of genuine reviews, while also incorporating irrelevant information that makes it more difficult to predict the product category.

In e-commerce systems, to annotate the node labels (i.e., product categories), machine learning-based automated labeling systems are commonly utilized. Specifically, human annotators manually label a small set of examples, which is used as the training examples to the machine learning model. Subsequently, a machine learning model is trained on these manually labeled product samples to automatically assign categories to other products. Therefore, the systems rely on the information about the products, e.g., reviews of products and co-purchase relationships, to assign categories to products. However, due to the influence of the fraudsters, the noisy node features (i.e., fake product reviews) and noisy graph structure (i.e., co-purchase relationships between irrelevant products) may hinder the accurate assignment of the automated labeling systems, which leads to the noisy node label. To replicate this procedure, we selected 5 examples per category class, which is equivalent to manual labeling process. We then trained a GCN model, leveraging the node features, graph structure, and manually labeled nodes, to predict the true product categories. Consequently, our set of labeled nodes are composed of both manually labeled nodes and nodes

labeled using the GCN model. Importantly, the labels of unlabeled nodes were left unchanged and still represented their actual categories. The data generation code is also available at `https://anonymous.4open.science/r/FDGN-PRINGLE-4B31/`.

We again emphasize that while existing works primarily focus on the unrealistic noise scenario where graphs contain only a single type of noise, to the best of our knowledge, this is the first attempt to understand the noise scenario in the real-world applications. Furthermore, we propose new graph benchmark datasets that closely imitate a real-world e-commerce system containing malicious fraudsters, which incurs FDGN. We expect these datasets to foster practical research in noise-robust graph learning.

### D.3 BASELINES

We compare PRINGLE with a wide range of noise-robust graph learning methods, which includes feature noise-robust grah learning methods (i.e., AirGNN (Liu et al., 2021)), structure-noise robust graph learning methods (i.e., ProGNN (Jin et al., 2020), RSGNN (Dai et al., 2022), STABLE (Li et al., 2022a) and EvenNet (Lei et al., 2022)), and label noise-robust graph learning methods (i.e., NRGNN (Dai et al., 2021) and RTGNN (Qian et al., 2023)). We also consider WSGNN (Lao et al., 2022) that is a generative graph learning method utilizing variational inference technique.

The publicly available implementations of baselines can be found at the following URLs:

- **AirGNN** (Liu et al., 2021) : https://github.com/lxiaorui/AirGNN

- **ProGNN** (Jin et al., 2020) : https://github.com/ChandlerBang/Pro-GNN

- **RSGNN** (Dai et al., 2022) : https://github.com/EnyanDai/RSGNN

- **STABLE** (Li et al., 2022a) : https://github.com/likuanppd/STABLE

- **EvenNet** (Lei et al., 2022) : https://github.com/Leirunlin/EvenNet

- **NRGNN** (Dai et al., 2022) : https://github.com/EnyanDai/NRGNN

- **RTGNN** (Dai et al., 2022) : https://github.com/GhostQ99/RobustTrainingGNN

- **WSGNN** (Lao et al., 2022) : https://github.com/Thinklab-SJTU/WSGNN

### D.4 EVALUATION PROTOCOL

We mainly compare the robustness of PRINGLE and the baselines under both the synthetic and real-world feature-dependent graph-noise (FDGN). More details of generating FDGN is provided in Sec D.2. Additionally, we consider independent feature/structure/label noise, i.e., random feature noise, random structure noise, uniform label noise, and pair label noise following existing works (Liu et al., 2021; Dai et al., 2022; Li et al., 2022a; Qian et al., 2023). Specifically, for the feature noise (Liu et al., 2021), we sample a subset of nodes (i.e., 10%, 30%, and 50%) and randomly flip 0/1 value on each dimension of node features $\mathbf{X}_i$ from Bernoulli distribution with probability $p = \frac{1}{F} \sum_{i=1}^{F} \mathbf{X}_i$. For the structure noise, we adopt the random perturbation method that randomly injects non-connected node pairs into the graph (Li et al., 2022a). For the label noise, we generate uniform label noise and pair label noise following the existing works (Qian et al., 2023; Dai et al., 2021).

We conduct both the node classification and link prediction tasks. For node classification, we perform a random split of the nodes, dividing them into a 1:1:8 ratio for training, validation, and testing nodes. Once a model is trained on the training nodes, we use the model to predict the labels of the test nodes. Regarding link prediction, we partition the provided edges into a 7:3 ratio for training and testing edges. Additionally, we generate random negatives that are selected randomly from pairs that are not directly linked in the original graphs. After mode learning with the training edges, we predict the likelihood of the existence of each edge. This prediction is based on a dot-product or cosine similarity calculation between node pairs of test edges and their corresponding negative edges. To evaluate performance, we use Accuracy as the metric for node classification and Area Under the Curve (AUC) for link prediction.

Table 6: Hyperparameter settings on PRINGLE for Table 1.

| Dataset | Setting | lr | $\lambda_1$ | $\lambda_2$ | $\theta_1$ | $k$ | $\gamma$ |
|---|---|---|---|---|---|---|---|
| Cora | Clean | 0.01 | 0.003 | 0.003 | 0.1 | 300 | 1 |
| | FDGN-10% | 0.005 | 0.003 | 0.003 | 0.2 | 50 | 1 |
| | FDGN-30% | 0.001 | 0.003 | 0.003 | 0.2 | 100 | 1 |
| | FDGN-50% | 0.0005 | 30 | 0.003 | 0.3 | 50 | 1 |
| Citeseer | Clean | 0.0005 | 0.003 | 0.3 | 0.1 | 50 | 0 |
| | FDGN-10% | 0.005 | 0.3 | 0.003 | 0.3 | 10 | 0 |
| | FDGN-30% | 0.001 | 0.003 | 0.003 | 0.1 | 300 | 1 |
| | FDGN-50% | 0.001 | 0.003 | 0.003 | 0.1 | 300 | 1 |
| Photo | Clean | 0.01 | 0.03 | 0.3 | 0.1 | 10 | 0 |
| | FDGN-10% | 0.0005 | 0.03 | 0.3 | 0.1 | 10 | 0 |
| | FDGN-30% | 0.001 | 3 | 0.03 | 0.1 | 10 | 0 |
| | FDGN-50% | 0.0005 | 30 | 0.03 | 0.1 | 10 | 0 |
| Comp | Clean | 0.01 | 30 | 0.03 | 0.1 | 10 | 0 |
| | FDGN-10% | 0.01 | 0.3 | 0.03 | 0.1 | 10 | 0 |
| | FDGN-30% | 0.01 | 0.003 | 0.003 | 0.1 | 10 | 0 |
| | FDGN-50% | 0.0005 | 0.003 | 0.03 | 0.1 | 10 | 0 |

### D.5 IMPLEMENTATION DETAILS

For each experiment, we report the average performance of 3 runs with standard deviations. For all baselines, we use the publicly available implementations and follow the implementation details presented in their original papers.

For PRINGLE, the learning rate is tuned from {0.01, 0.005, 0.001, 0.0005}, and dropout rate and weight decay are fixed to 0.6 and 0.0005, respectively. In the inference of $Z_A$, we use a 2-layer GCN model with 64 hidden dimension as $\text{GCN}_{\phi_1}$ and the dimension of node embedding $d_1$ is fixed to 64. The $\gamma$ value in calculating $\gamma$-hop subgraph similarity is tuned from {0, 1} and $k$ in generating $k$-NN graph is tuned from {0, 10, 50, 100, 300}. In the inference of $Z_Y$, we use a 2-layer GCN model with 128 hidden dimension as $\text{GCN}_{\phi_3}$. In the inference of $\epsilon_X$, the hidden dimension of embedding of $\epsilon_X$ is fixed to 16. In the inference of $\epsilon_A$, the early-learning phase is fixed to 30 epochs. In the implementation of the loss term $-\mathbb{E}_{Z_A \sim q_{\phi_1}} \mathbb{E}_{\epsilon \sim q_{\phi_2}} [\log(p_{\theta_1}(A|X, \epsilon, Z_A))]$, we tune the $\theta_1$ value from {0.1, 0.2, 0.3}. In the overall learning objective, i.e., Eqn 4, $\lambda_1$ is tuned from { 0.003, 0.03. 0.3, 3, 30 }, $\lambda_2$ is tuned from { 0.003, 0.03. 0.3 }, and $\lambda_3$ is fixed to 0.001. It is important to note that when we calculate Eqn. 4, $\mathcal{L}_{\text{rec-feat}}$, $\mathcal{L}_{\text{cls-dec}}$, and $\mathcal{L}_{\text{p}}$ share the same coefficient $\lambda_3$. In our experiments, we observed that these three terms have a relatively minor impact on the model's performance compared to the others. As a result, we have made a strategic decision to simplify the hyperparameter search process and improve the practicality of PRINGLE by sharing the coefficient $\lambda_3$ among these three loss terms. We report the details of hyperparameter settings in Table 6.

## E ADDITIONAL EXPERIMENTAL RESULTS

### E.1 UNDER INDEPENDENT FEATURE/STRUCTURE/LABEL NOISE SETTINGS

In this subsection, we further evaluate the robustness of PRINGLE under independent feature/structure/label noise settings. In this setup, each type of noise occurs independently and does not affect the occurrence of the others. To this end, we generate three types of noise: random feature noise, random structure noise, and random label noise. More details of generating noises is provided in Sec D.4.

**Evaluating robustness under independent feature noise.** In this setting, we evaluate the models on a graph containing only the feature noise, i.e., random feature noise. In Fig 6, PRINGLE consistently outperforms the feature noise-robust graph learning method (i.e., AirGNN) under independent feature noise. We attribute the robustness of PRINGLE under independent feature noise to the graph structure learning module that accurately infers the latent graph structure $Z_A$. The utilization of abundant local neighborhoods acquired through the inference of $Z_A$ enables effective smoothing for nodes with noisy features, leveraging the information within these neighborhoods.

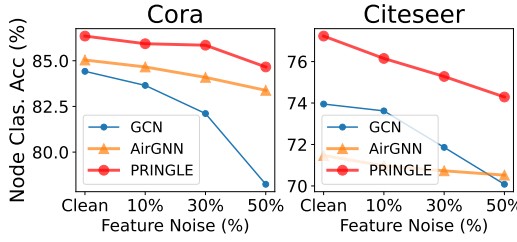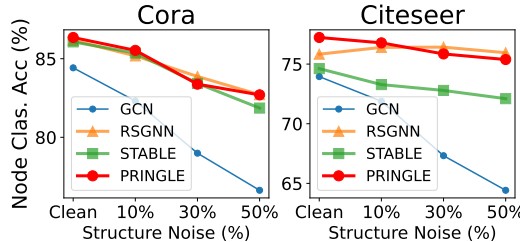

Figure 6: Node classification performance of baselines and PRINGLE on Cora and Citeseer dataset under independent feature noise. We vary a noise rate from 0% to 50%

Figure 7: Node classification performance of baselines and PRINGLE on Cora and Citeseer dataset under independent structure noise. We vary a noise rate from 0% to 50%.

**Evaluating robustness under independent structure noise.** In this setting, we evaluate the models on a graph containing only the structure noise, i.e., random structure noise. In Fig 7, Under the influence of independent structure noise, PRINGLE maintains a consistently competitive performance when compared to other structure noise-robust graph learning methods, namely RSGNN and STABLE. We attribute the effectiveness of PRINGLE under independent structure noise to inferring the robust latent clean graph structure. In other words, the inference of the latent clean graph structure $Z_A$ assigns greater weights to latent clean edges and lower weights to observed noisy edges by employing regularizations on both the edge predictions and labels, thereby mitigating structural noise.

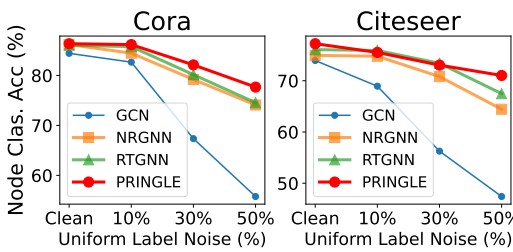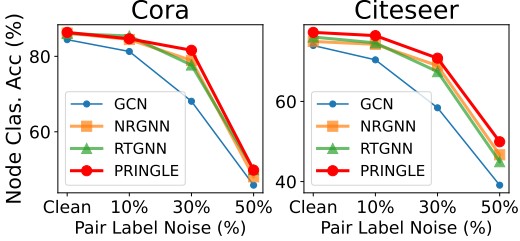

Figure 8: Node classification performance of baselines and PRINGLE on Cora and Citeseer dataset under independent uniform label noise. We vary a noise rate from 0% to 50%.

Figure 9: Node classification performance of baselines and PRINGLE on Cora and Citeseer dataset under independent pair label noise. We vary a noise rate from 0% to 50%.

**Evaluating robustness under independent label noise** In this setting, we evaluate the models on a graph containing only the label noise, i.e., uniform label noise and pair label noise. In Fig 8 and 9, PRINGLE demonstrates consistent superiority or competitive performance compared to label noise-robust graph learning methods, namely NRGNN and RTGNN, in the presence of independent label noise. We argue that the effectiveness of PRINGLE stems from the accurate inference of the latent clean structure. Specifically, the inferred latent node label $Z_Y$ is regularized using the inferred latent structure $Z_A$ to meet the homophily assumption (i.e., $\mathcal{L}_{\text{hom}}$). Leveraging the clean neighbor structure, this regularization technique has been demonstrated to effectively address noisy labels (Iscen et al., 2022).

**Evaluating robustness under simultaneous noise.** In addition to a single type of independent noise, we explore a more challenging yet realistic noise scenario where all three types of independent noises occur simultaneously, denoted as simultaneous noise. More specifically, for the feature and structure noise, we generate random feature noise and random structure noise in a same way as described in Sec D.4. Regarding the label noise, we generate uniform label noise following Qian et al. (2023). It is important to note that each type of noise does not affect the occurrence of the other types of noise. In Fig 10, PRINGLE consistently outperforms the noise-robust graph learning methods and generative graph learning methods under simultaneous noise setting. Based on these results, we can assert that modeling the DGP of FDGN is advantageous for robustness under independent graph noise, as PRINGLE is inherently capable of handling each type of noise. In contrast, the baseline methods

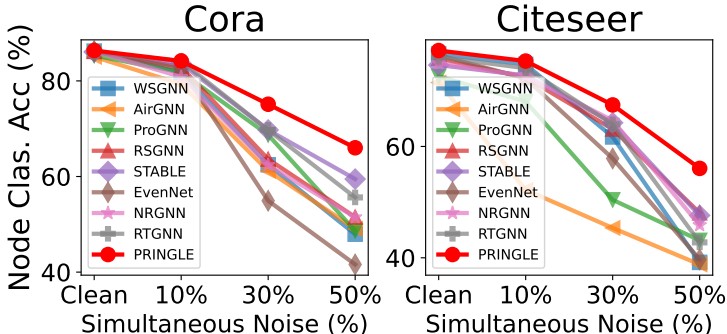

Figure 10: Node classification performance of baselines and PRINGLE on Cora and Citeseer dataset under simultaneous noise, where random feature, structure, and label noise independently exist in a graph. We vary a noise rate from 0% to 50%.

Table 7: Comparison with the naive combination of existing noise-robust graph learning methods. *FNR*, *SNR*, and *LNR* denote the feature noise-robust, structure noise-robust, and label noise-robust graph learning methods, respectively. We consider AirGNN as *FNR*, RSGNN as *SNR*, and RTGNN as *LSR* methods.

| Component | | | Cora | | | | Citeseer | | | |
|---|---|---|---|---|---|---|---|---|---|---|
| *FNR* | *SNR* | *LNR* | Clean | FDGN 10% | FDGN 30% | FDGN 50% | Clean | FDGN 10% | FDGN 30% | FDGN 50% |
| ✓ | ✗ | ✗ | 85.0±0.2 | 79.7±0.5 | 71.5±0.8 | 56.2±0.8 | 71.5±0.2 | 66.2±0.7 | 58.0±0.4 | 50.0±0.6 |
| ✗ | ✓ | ✗ | 86.2±0.5 | 81.9±0.3 | 71.9±0.5 | 58.1±0.2 | 75.8±0.4 | 73.3±0.5 | 63.9±0.5 | 55.3±0.4 |
| ✗ | ✗ | ✓ | 86.1±0.2 | 81.6±0.5 | 72.1±0.6 | 60.8±0.4 | 76.1±0.4 | 73.2±0.2 | 63.5±2.1 | 54.2±1.8 |
| ✓ | ✓ | ✗ | 86.0±0.3 | 82.0±0.3 | 75.0±0.8 | 68.8±0.6 | 75.1±0.8 | 73.1±0.6 | 63.6±0.8 | 57.8±0.8 |
| ✓ | ✗ | ✓ | 85.2±0.7 | 70.1±0.1 | 56.7±0.4 | 48.0±0.5 | 75.8±0.5 | 72.3±0.3 | 59.0±0.7 | 49.0±0.2 |
| ✗ | ✓ | ✓ | 85.0±0.2 | 79.4±0.9 | 72.3±0.5 | 63.0±0.4 | 76.7±0.3 | **74.3±0.9** | 64.8±0.3 | 55.3±0.5 |
| ✓ | ✓ | ✓ | **86.3±0.3** | 82.4±0.3 | 67.0±0.9 | 53.6±0.6 | 76.6±0.2 | 73.0±0.7 | 64.1±0.2 | 52.7±1.1 |
| PRINGLE | | | 86.2±0.7 | **82.9±0.6** | **78.2±0.3** | **69.7±0.6** | **77.3±0.6** | **74.3±0.9** | **65.6±0.6** | **59.0±1.8** |

assume the completeness of at least one of the data sources, resulting in a significant performance drop when the noise rate is high. This suggests that PRINGLE has a broader range of applicability in real-world scenarios.

### E.2 COMPARISON WITH THE NAIVE COMBINATION OF EXISTING WORKS

So far, we have observed that existing approaches fail to generalize to FDGN since they primarily focus on graphs containing only a single type of noise. A straightforward solution might be to naively combine methods that address each type of noise individually. To explore this idea, we consider AirGNN as the feature noise-robust graph learning method (*FNR*), RSGNN as the structure noise-robust graph learning method (*SNR*), and RTNN as the label noise-robust graph learning method (*LNR*). We carefully implement all possible combinations among *FNR*, *SNR*, and *LNR*.

In Table 7, we observe that naive combination can improve robustness in some cases, but it may not consistently yield favorable results. For example, combining *FNR* and *SNR* notably enhances robustness. However, when we combine all three (*FNR*, *SNR*, and *LNR*), which is expected to yield the best results, performance even decreases. This could be attributed to compatibility issues among the methods arising from the naive combination. Furthermore, although some combinations improve robustness, PRINGLE consistently outperforms all combinations. We attribute this to the fact that naively combining existing methods may not capture the causal relationships in the DGP of FDGN, limiting their robustness. In contrast, PRINGLE successfully captures these relationships, resulting in superior performance.

### E.3 SENSITIVITY ANALYSIS

In this section, we analyze the sensitivity of the coefficient $\lambda_1$, $\lambda_2$, and $\lambda_3$ in Eqn 4. To be specific, we increase $\lambda_1$ value from $\{0.0, 0.003, 0.03, 0.3, 3\}$, $\lambda_2$ value from $\{0.0, 0.003, 0.03, 0.3\}$, and $\lambda_3$ from $\{0.0, 0.01, 0.1, 1, 10\}$. We then evaluate the node classification accuracy of PRINGLE under FDGN.

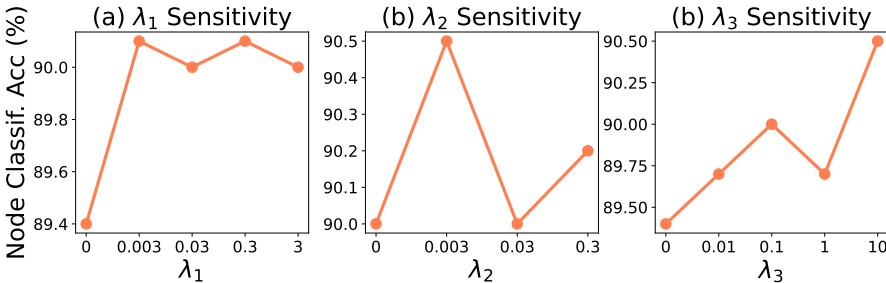

Figure 11: Sensitivity analysis on $\lambda_1$, $\lambda_2$, and $\lambda_3$. We conduct the experiments on Photo dataset under FDGN-30%

In Fig 11(a), we observe that the performance significantly drops when $\lambda_1 = 0$. This highlights the importance of modeling the causal relationship $A \leftarrow (X, \epsilon, Z_A)$ for robustness under FDGN, as $\lambda_1$ is directly related to the loss term $\mathcal{L}_{\text{edge-rec}}$, i.e., $-\mathbb{E}_{Z_A \sim q_{\phi_1}} \mathbb{E}_{\epsilon \sim q_{\phi_2}} [\log(p_{\theta_1}(A|X, \epsilon, Z_A))]$.

In Fig 11(b), we can see a performance decrease when $\lambda_2 = 0$. This observation suggests that the regularization on the inferred latent node label $Z_Y$ using the inferred latent structure $Z_A$ effectively handles the noisy labels. This conclusion is drawn from the fact that $\lambda_2$ is directly linked to the loss term $\mathcal{L}_{\text{hom}}$, i.e., $kl(q_{\phi_3}(Z_Y|X, A)||p(Z_Y))$.

In Fig 11(c), we can clearly observe a substantial performance drop when $\lambda_3 = 0$, and the performance tends to improve as $\lambda_3$ increases. This observation strongly suggests the importance of modeling the causal relationships $X \leftarrow (\epsilon, Z_Y)$ and $Y \leftarrow (X, A, Z_Y)$ for robustness under FDGN. We draw this conclusion by considering that $\lambda_3$ is directly associated to the loss terms $\mathcal{L}_{\text{rec-feat}}$, $\mathcal{L}_{\text{cls-dec}}$, and $\mathcal{L}_{\text{p}}$, i.e., $-\mathbb{E}_{\epsilon \sim q_{\phi_2}} \mathbb{E}_{Z_Y \sim q_{\phi_3}} [\log(p_{\theta_2}(X|\epsilon, Z_Y))]$, $-\mathbb{E}_{Z_Y \sim q_{\phi_3}} [\log(p_{\theta_3}(Y|X, A, Z_Y))]$, and $\mathbb{E}_{Z_Y \sim q_{\phi_3}} [kl(q_{\phi_2}(\epsilon|X, A, Z_Y)||p(\epsilon))]$.

### E.4 ANALYSIS OF THE INFERRED $Z_A$ AND $\epsilon_A$

In this subsection, we qualitatively analyze how well PRINGLE infers the latent variable $\epsilon_A$ and $Z_A$. In Fig 12(a), we conducted an analysis of the inference of $\epsilon_A$ by comparing the distribution of $\hat{p}_{ij}^{el}$ values estimated during the training of PRINGLE on clean and noisy graphs (FDGN-50%). We observe that $\hat{p}_{ij}^{el}$ values estimated from the clean graph tend to be close to 1, while those from the graph with FDGN are considerably smaller. This observation suggests that the inference of $\epsilon_A$ was accurate, as the high values of $\hat{p}_{ij}^{el}$ indicate that the model recognizes the edge $(i, j)$ as a clean edge.

In Fig 12(b), we analyze the inference of $Z_A$ by comparing the distribution of $\hat{p}_{ij}$ values, which constitute the estimated latent graph structure $\hat{\mathbf{A}}$, between noisy edges and the original clean edges. It is evident that the estimated edge probabilities $\hat{p}_{ij}$ for noisy edges are predominantly assigned smaller values, while those for clean edges tend to be assigned larger values. This observation illustrates that PRINGLE effectively mitigates the impact of noisy edges during the message-passing process, thereby enhancing its robustness in the presence of noisy graph structure.

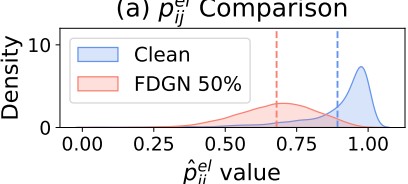

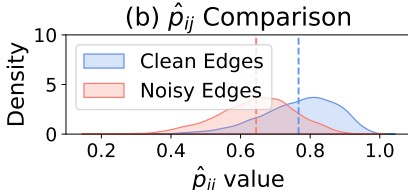

Figure 12: (a) Distribution of $\hat{p}_{ij}^{el}$ values estimated from clean and FDGN-50% Cora dataset. (b) Distribution of $\hat{p}_{ij}$ values of clean edges and noisy edges under FDGN-50%. Dashed lines indicate average values. Cora dataset is used.

This achievement can be attributed to the label regularization effect achieved through the accurate inference of $\epsilon_A$. Specifically, as the observed graph structure contains noisy edges, the inaccurate supervision for $\mathcal{L}_{\text{rec-edge}}$ impedes the distinction between noisy edges and the original clean edges in terms of edge probability values $\hat{p}_{ij}$. However, the label regularization technique proves crucial for alleviating this issue, benefitting from the accurate inference of $\epsilon_A$.

## F   FURTHER DISCUSSION ON FDGN

In this work, our main focus is on the limitations of the conventional graph noise assumption in terms of node features: the noise in node features is independent of the graph structure or node labels. To this end, we propose a more realistic noise scenario FDGN by introducing the causal relationships $A \leftarrow X$ and $Y \leftarrow (X, A)$ into the DGP of CGN. However, it is crucial to acknowledge that in some real-world scenarios, the causal relationship $X \leftarrow A$ may indeed manifest, which indicates that the graph structure noise inevitably entails the node feature noise. For instance, consider a social network where node features represent the content to which a user is exposed or interacts with (e.g., views, clicks, or likes), while the graph structure denotes the follower relationships. In such a scenario, if a user follows or is followed by fake accounts, the graph structure might incorporate noisy links (i.e., noisy graph structure). This, in turn, can impact the content to which users are exposed and their

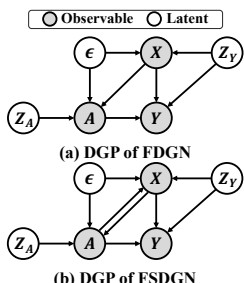

Figure 13: A directed graphical model indicating a DGP of (a) FDGN, and (b) FSDGN.

interactions (i.e., noisy node features), eventually influencing their community assignments (i.e., noisy node labels). In other words, the noisy node feature and noisy graph structure mutually influence the noise of each other, ultimately incurring the noisy node label. We denote this scenario as **f**eature **s**tructure-**d**ependent **g**raph-**n**oise (FSDGN), and its DGP is illustrated in Fig 13(b). Given that the DGP of FSDGN covers a broader range of noise scenarios that occur in real-world applications than FDGN, we expect that directly modeling the DGP of FSDGN has the potential to enhance practical applicability. However, this is a topic we leave for future work.

## G   COMPARISON TO CAUSALNL (YAO ET AL., 2021)

One can suggest that our work may appear to lack technical novelty in comparison to CausalNL (Yao et al., 2021). However, it is important to note that we tackle additional, more challenging aspects unique to our specific problem that do not exist in CausalNL. Specifically, when additionally introducing $\mathbf{A}$, we handle four more causal relationships: $A \leftarrow \epsilon$, $A \leftarrow X$, $Y \leftarrow A$, $A \leftarrow Z_A$, each of which is non-trivial to consider. We would like to specify the challenges in instantiations caused by their additional introduction.

**Inference of $\epsilon$.** In contrast to CausalNL, which assumes that $\epsilon$ is a cause of $X$, our proposed FDGN consider that $\epsilon$ is a cause of both $X$ and $A$. In other words, the DGP of FDGN contains the causal relationships $X \leftarrow \epsilon$ and $A \leftarrow \epsilon$, as real-world applications often exhibit graph structure noise originating from arbitrary sources (i.e., $\epsilon$) in addition to the feature-dependent noise. Therefore, this scenario is a unique characteristic of our problem and not addressed in CausalNL. To deal with this, we decompose $q_{\phi_2}(\epsilon|X, A, Z_Y)$ into $q_{\phi_{21}}(\epsilon_X|X, Z_Y)$ and $q_{\phi_{22}}(\epsilon_A|X, A)$. While the instantiation of $q_{\phi_{21}}(\epsilon_X|X, Z_Y)$ is similar to CausalNL, that of $q_{\phi_{22}}(\epsilon_A|X, A)$ is non-trivial and is absent in CausalNL. In our approach, we regard $\epsilon_A$ as a set of scores indicating the likelihood of each observed edge being noisy or not. Moreover, we leverage the concept of early-learning phenomenon to infer $\epsilon_A$. It is worth emphasizing once again that the instantiation of this scenario is novel and not a straightforward extension of CausalNL.

**Loss term $kl(q_{\phi_1}(Z_A|X, A)||p(Z_A))$**  To compute this loss term, we encounter two primary challenges:

1. Designing an appropriate prior for the latent graph structure $p(Z_A)$
2. Addressing the complexity associated with calculating the KL divergence between the two matrices sampled from Bernoulli distributions.

In response to the first challenge, we employ the $\gamma$-hop subgraph similarity as a metric to identify assortative edges. Regarding the second challenge, we introduce a predefined candidate graph, which includes the observed edge set along with a $k$-NN graph based on the $\gamma$-hop subgraph similarity. Both of these challenges represent non-trivial aspects of our approach and are absent in CausalNL.

**Loss term $kl(q_{\phi_3}(Z_Y|X, A)||p(Z_Y))$**  To compute this loss term, CausalNL employs a uniform distribution as the prior $p(Z_Y)$. In contrast, we introduce the concept of class homophily to effectively

regularize the inference of latent clean node label $Z_Y$. Specifically, we encourage $Z_Y$ to align with our prior knowledge, i.e., $p(Z_Y)$, that the two end nodes on the accurately inferred latent graph structure $Z_A$ are expected to have identical latent labels. Therefore, using this prior helps alleviate the noisy node label issues. It is essential to highlight that this instantiation effectively utilizes the property unique to the graph domain, which is not present in CausalNL. It is worth emphasizing once again that such an instantiation is novel and not a straightforward extension of CausalNL.

**Loss term** $\mathbb{E}_{Z_Y \sim q_{\phi_3}}[kl(q_{\phi_2}(\epsilon|X, A, Z_Y)||p(\epsilon))]$     This term is decomposed into $kl(q_{\phi_{21}}(\epsilon_X|X, Z_Y)||p(\epsilon_X))$ and $kl(q_{\phi_{22}}(\epsilon_A|X, A)||p(\epsilon_A))$. The first term is calculated in a similar manner to CausalNL. Specifically, a Gaussian distribution is employed as the prior $p(\epsilon_X)$. However, the second term is unique to our problem, and its computation necessitates prior information about $\epsilon_A$. As $\epsilon_A$ indicates the likelihood of each observed edge being noisy or not, it is not appropriate to simply assume $p(\epsilon_A)$ as a Gaussian distribution as done in CausalNL. This aspect makes the problem more challenging. To address this challenge, we introduce a new assumption that the inferred $\epsilon_A$ follows an unknown distribution with high variance, while our prior knowledge suggests that $p(\epsilon_A)$ follows the same distribution but with low variance. Additionally, we employ an Exponential Moving Average (EMA) technique to reduce the uncertainty of the inferred $\epsilon_A$. This challenge represent non-trivial aspects of our approach and is absent in CausalNL.

## H    ADDITIONAL REAL-WORLD EXAMPLES OF FDGN

In this paper, we consistently assert that FDGN represents a more realistic graph noise scenario. In order to substantiate this assertion further, we present additional real-world application examples, in addition to those involving social networks and co-purchase networks:

- **Biological Networks**: A cell-cell graph is widely used in computational biology. In the cell-cell graph, each node represents each cell. Then corresponding node features and node labels are represented as gene expression and cell type, respectively. However, gene expression as cell-gene count matrix often contain noises such as dropout phenomenon and batch effect. Due to the lack of cell-cell graph structure (i.e., graph adjacency), many works design cell-cell graph structures via utilizing the node features (Wang et al., 2021; Xiong et al., 2023; Yun et al., 2023), which may entail the noisy cell-cell graph structures. Furthermore, regarding the cell type (i.e., node label) is annotated by using transcripted marker genes (i.e., important features), noisy node features may lead the noisy node labels.

- **Recommendation systems**: In recommendation systems of various domains (e.g., e-commerce, news, and music), user-item interaction graphs are common. The node features may represent users' and products' information and user-item interactions represent the graph structures. Furthermore, the node labels can be represented by the users' communities (interests) and products' categories. When a user creates a fake or incomplete profile due to various reasons (e.g., privacy), products irrelevant to the user's genuine interest can be exposed in a web/app page to promote the user to view/click/purchase the products. Then, the user is more likely to interact with irrelevant products (i.e., noisy graph structure) due to the user's noisy features. Furthermore, such noisy users' information and interactions may also eventually change their communities (i.e., noisy node label).

## I    FURTHER COMPARISON TO LABEL NOISE-ROBUST BASELINES

We agree with the reviewer's comment about the missing baselines. Hence, we further compare PRINGLE with the widely used label noise baselines: Co-teaching+ (Yu et al., 2019), CP (Zhang et al., 2020), D-GNN (NT et al., 2019), and CGNN (Yuan et al., 2023). From Table 8, we clearly see that the proposed method, PRINGLE, outperforms all the baselines. We argue that these baseline methods were designed to tackle the noisy node labels while assuming that both node features and graph structure are noise-free. However, the proposed FDGN introduces a more realistic noise scenario, wherein node features, graph structures, and node labels simultaneously contain noise. As a consequence, the performance of these existing methods is considerably more constrained compared to PRINGLE.

Table 8: Node classification performance under synthetic feature-dependent graph-noise (FDGN).

| Dataset | Setting | Co-teaching+ | CP | D-GNN | CGNN | PRINGLE |
|---------|---------|--------------|-----|-------|------|---------|
| Cora | Clean | 84.7±0.9 | 84.3±0.4 | 83.7±0.5 | 85.2±0.7 | **86.2±0.7** |
| | FDGN-10% | 76.9±0.5 | 78.7±0.6 | 78.6±0.5 | 77.4±0.3 | **82.9±0.6** |
| | FDGN-30% | 66.0±0.8 | 68.2±0.5 | 68.5±0.4 | 69.2±0.8 | **78.2±0.3** |
| | FDGN-50% | 55.0±0.1 | 53.1±0.9 | 57.7±0.3 | 55.1±0.2 | **69.7±0.6** |
| Citeseer | Clean | 72.7±0.4 | 72.8±0.9 | 75.1±0.2 | 71.1±0.9 | **77.3±0.6** |
| | FDGN-10% | 67.7±0.6 | 68.4±0.8 | 69.0±0.5 | 65.6±0.4 | **74.3±0.9** |
| | FDGN-30% | 55.0±0.9 | 56.8±0.9 | 54.0±0.7 | 54.1±0.3 | **65.6±0.6** |
| | FDGN-50% | 47.4±0.8 | 46.5±1.0 | 44.7±0.1 | 46.9±1.4 | **59.0±1.8** |
| Photo | Clean | 93.1±0.0 | 93.3±0.5 | 93.1±0.1 | 92.7±0.5 | **94.8±0.3** |
| | FDGN-10% | 87.9±0.8 | 90.5±0.5 | 90.3±0.6 | 87.1±0.3 | **93.2±0.2** |
| | FDGN-30% | 83.1±0.2 | 85.1±1.0 | 85.9±0.3 | 85.1±0.2 | **90.5±0.4** |
| | FDGN-50% | 61.9±0.3 | 80.4±0.6 | 85.1±0.8 | 80.5±0.7 | **87.6±0.2** |
| Comp | Clean | 88.6±0.8 | 90.7±0.3 | 89.4±0.9 | 90.0±0.5 | **92.2±0.0** |
| | FDGN-10% | 85.6±0.6 | 87.1±0.8 | 86.8±0.4 | 83.0±0.4 | **89.8±0.2** |
| | FDGN-30% | 81.5±0.3 | 82.8±0.6 | 82.9±0.5 | 82.3±0.8 | **86.9±0.3** |
| | FDGN-50% | 72.8±0.9 | 74.3±1.0 | 74.5±0.6 | 75.3±0.1 | **82.2±0.4** |

## J    COMPLEXITY ANALYSIS

Table 9: Training time comparison on Cora dataset under FDGN 50%.

| | WSGNN | AirGNN | ProGNN | RSGNN | STABLE | EvenNet | NRGNN | RTGNN | PRINGLE |
|---|-------|--------|--------|-------|--------|---------|-------|-------|---------|
| Total training time (sec) | 93.90 | 20.9 | 702.14 | 159.87 | 53.33 | **0.81** | 100.33 | 118.7 | 46.27 |
| Training time / epoch (sec) | 0.19 | 0.04 | 1.77 | 0.16 | - | **0.004** | 0.20 | 0.18 | 0.09 |

We compare the training time of PRINGLE with the baselines to analyze the computational complexity of PRINGLE . In Table 9, we report the total training time and training time per epoch on Cora with FDGN 50% for all models. Note that since STABLE is a 2-stage method, we did not report the training time per epoch. The results show that PRINGLE requires significantly less total training time and training time per epoch compared to WSGNN, ProGNN, RSGNN, STABLE, NRGNN, and RTGNN. This suggests that PRINGLE's training procedure is faster than that of most baselines while still achieving substantial performance improvements. Although AirGNN and EvenNet require much less training time than PRINGLE, their node classification accuracy is notably worse than other methods, including PRINGLE. This indicates that, despite their fast training times, they may not be suitable for real-world scenarios. In summary, PRINGLE demonstrates superior performance compared to the baselines while maintaining acceptable training times.

## K    FURTHER DISCUSSION WITH GRAPHDE AND RGIB

Li et al. (2022b) introduced a debiased learning framework, GraphDE, designed to handle situations where out-of-distribution (OOD) samples are present in training data. Given that a noisy sample can be viewed as an OOD sample, GraphDE shares a similar objective with our work. Additionally, GraphDE employs a generative model based on variational inference, relevant to learning approach. However, there are significant distinctions in the DGP that we assume compared to GraphDE. Specifically, GraphDE assumes that a latent variable $e$ determines whether an instance is an OOD sample or not, overlooking the way that the OOD sample is generated. Conversely, the DGP of our proposed FDGN characterizes the process by which noisy samples (i.e., OOD sample) are generated (e.g., in a feature-dependent manner). This crucial difference enables a more fine-grained model learning than GraphDE, which may weaken the applicability of GraphDE to complex real-world noise scenarios, such as FDGN.

Zhou et al. (2023) introduced a graph structure denoising framework called RGIB, primarily centered on the link prediction task. In their work, while $Z_A$ and $A$ share the same meaning as in our paper, $Y$ denotes "edge labels" rather than node labels (i.e., it serves as a binary indicator for the presence of query edges), and $Z_Y$ refers to "clean edge labels" rather than clean latent node labels. These distinctions significantly differentiate their approach from ours. Furthermore, it's important to note that RGIB primarily addressed noisy graph structures while assuming that node features and node labels are noise-free. In contrast, our proposed method, PRINGLE, is developed under the more realistic FDGN assumption, where node features, graph structures, and node labels all simultaneously contain noise. This fundamental difference underscores the enhanced applicability and robustness of PRINGLE compared to RGIB.

---

**Algorithm 1** Training Algorithm of PRINGLE.

---

1: **Input**: Observed graph $\mathcal{G} = \langle \mathcal{V}, \mathcal{E} \rangle$, node feature $\mathbf{X} \in \mathbb{R}^{N \times F}$, node label $\mathbf{Y} \in \mathbb{R}^{N \times C}$
2: Initialize trainable parameters $\phi_1, \phi_2, \phi_3, \theta_2, \theta_3$
3: Initialize $\hat{p}_{ij}^{el}$ to one vector $\mathbf{1}$.
4: Generate a $k$-NN graph $\mathcal{E}_k^{\gamma}$ based on the $\gamma$-hop subgraph similarity
5: Pre-define a candidate graph by $\mathcal{E}_k^{\gamma} \cup \mathcal{E}$
6: **while** *not converge* **do**
7:     /* Inference of $Z_A$ */
8:     Feed $\mathbf{X}$ and $\mathbf{A}$ to GCN$_{\phi_1}$ to obtain the node embeddings $\mathbf{Z}$
9:     Calculate the $\hat{p}_{ij}$ on the candidate graph $\mathcal{E}_k^{\gamma} \cup \mathcal{E}$ based on $\mathbf{Z}$ to obtain $\hat{\mathbf{A}}$.
10:     /* Inference of $Z_Y$ */
11:     Feed $\mathbf{X}$ and $\hat{\mathbf{A}}$ to GCN$_{\phi_3}$ to get $\hat{\mathbf{Y}}$
12:     /* Inference of $\epsilon_X$ */
13:     Feed $\mathbf{X}$ and $\hat{\mathbf{Y}}$ to the MLP$_{\phi_2}$ to get node embeddings that follow $\mathcal{N}(\mathbf{0}, \mathbf{I})$
14:     /* Inference of $\epsilon_A$ */
15:     **if** early-learning phase **then**
16:         $\hat{p}_{ij}^c \leftarrow \rho(s(\mathbf{Z}_i, \mathbf{Z}_j))$
17:         $\hat{p}_{ij}^{el} \leftarrow \xi \hat{p}_{ij}^{el} + (1 - \xi) \hat{p}_{ij}^c$
18:         Convert $\hat{p}_{ij}^{el}$ into $\tau_{ij}$
19:     **end if**
20:     /* Generation of $A$ */
21:     Obtain an edge prediction $w_{ij} = \theta_1 \hat{p}_{ij} + (1 - \theta_1) s(\mathbf{X}_i, \mathbf{X}_j)$
22:     /* Generation of $X$ */
23:     Obtain the reconstruction of node features based on decoder MLP$_{\theta_2}$ and its input $\epsilon_X$ and $\hat{\mathbf{Y}}$.
24:     /* Generation of $Y$ */
25:     Obtain node prediction $\hat{\mathbf{Y}}_{\text{dec}}$ based on classifier GCN$_{\theta_3}$ and its input $\mathbf{X}$ and $\mathbf{A}$.
26:     /* Loss calculation */
27:     Calculate the objective function $\mathcal{L}_{\text{cls-enc}} + \lambda_1 \mathcal{L}_{\text{rec-edge}} + \lambda_2 \mathcal{L}_{\text{hom}} + \lambda_3 (\mathcal{L}_{\text{rec-feat}} + \mathcal{L}_{\text{cls-dec}} + \mathcal{L}_{\text{p}})$.
28:     /* Parameter updates */
29:     Update the parameters $\phi_1, \phi_2, \phi_3, \theta_2, \theta_3$ to minimize the overall objective function.
30: **end while**
31: **Return:** learned model parameters $\phi_1, \phi_2, \phi_3, \theta_2, \theta_3$

---

---

**Algorithm 2** Data Generation Algorithm of Synthetic FDGN.

---

1: **Input**: Clean graph $\mathcal{G} = \langle \mathcal{V}, \mathcal{E} \rangle$, node feature $\mathbf{X} \in \mathbb{R}^{N \times F}$, node label $\mathbf{Y} \in \mathbb{R}^{N \times C}$, noise rate $\eta\%$
2: /* Injection of feature noise */
3: $\mathcal{V}^{\text{noisy}} \leftarrow$ Randomly sample a $\eta\%$ subset of nodes
4: $\mathbf{X}^{\text{noisy}} \leftarrow \mathbf{X}$
5: **for** $v_i$ in $\mathcal{V}^{\text{noisy}}$ **do**
6: $\quad p_i \leftarrow \frac{1}{F} \sum_{j=1}^{F} \mathbf{X}_{ij}$
7: $\quad$ **for** $j \leftarrow 1$ to $F$ **do**
8: $\quad\quad \mathbf{X}_{ij}^{\text{noisy}} \leftarrow$ **BernoulliSample**$(p_i)$
9: $\quad$ **end for**
10: **end for**
11: /* Injection of feature-dependent structure noise */
12: $\mathcal{E}^{\text{noisy}} \leftarrow \mathcal{E}$
13: **for** $v_i$ in $\mathcal{V}^{\text{noisy}}$ **do**
14: $\quad \mathbf{s} \leftarrow \mathbf{0} \in \mathbb{R}^{N}$
15: $\quad$ **for** $j \leftarrow 1$ to $N$ **do**
16: $\quad\quad \mathbf{s}_j \leftarrow s(\mathbf{X}_i^{\text{noisy}}, \mathbf{X}_j)$
17: $\quad$ **end for**
18: $\quad$ Append $k$ pairs of nodes with the highest $\mathbf{s}$ values to $\mathcal{E}^{\text{noisy}}$
19: **end for**
20: /* Injection of feature-dependent label noise */
21: $\mathbf{Y}^{\text{noisy}} \leftarrow \mathbf{Y}$
22: **for** $v_i$ in $\mathcal{V}^L$ **do**
23: $\quad$ **if** $v_i$ has noisy feature or noisy structure **then**
24: $\quad\quad \mathbf{p}_i \leftarrow$ Obtain normalized neighborhood class distribution of node $v_i$
25: $\quad\quad \mathbf{Y}_i^{\text{noisy}} \leftarrow$ **MultinomialSample**$(\mathbf{p}_i)$
26: $\quad$ **end if**
27: **end for**
28: /* Injection of independent structure noise */
29: Randomly append pairs of nodes to $\mathcal{E}^{\text{noisy}}$
30: **Return:** noisy graph $\mathcal{G} = \langle \mathcal{V}, \mathcal{E}^{\text{noisy}} \rangle$, noisy node feature $\mathbf{X}^{\text{noisy}}$, noisy node label $\mathbf{Y}^{\text{noisy}}$

---

