# OpenReview forum: "Noise Robust Graph Learning under Feature-Dependent Graph-Noise"
_ICLR.cc/2024/Conference — Submitted to ICLR 2024_

### Official Review · Reviewer_7Duj · 2023-10-30

**Soundness:** 4 excellent
**Presentation:** 4 excellent
**Contribution:** 4 excellent
**Rating:** 8
**Confidence:** 4

**Summary:**

This paper focuses on the problem of node feature noise in graph learning. In this paper, the author claims that existing methods make an unrealistic assumption that the noise in the node features is independent of the graph structure or node labels, while a more realistic assumption should be that noisy node features may entail both structure and label noise. Under such an assumption, this paper proposes a principled noisy graph learning framework named PRINGLE to address the feature noise problem in graph learning. Experimental results based on several datasets are reported.

**Strengths:**

-	The problem of feature noise in graph learning is an important problem.
-	To the best of my knowledge, the assumption that noisy node features may entail both structure and label noise is novel, and this paper provides examples and empirical evidence to show that such an assumption is realistic.
-	The proposed PRINGLE method includes a deep generative model that directly models the data-generating process of the feature-dependent graph noise to capture the relationship among the variables that introduce noise. The proposed PRINGLE method generally makes sense.
-	Empirical evidence based on both existing benchmark datasets and newly collected datasets has been provided to show that PRINGLE outperforms state-of-the-art baselines in addressing the feature-dependent graph noise problem.

**Weaknesses:**

-	Minor issues about the typo: “the graph structure OF node labels” in line 4 of the abstract should be “the graph structure OR node labels” if I am not misunderstanding. Besides, in line 5 of page 5, “introduces” should be “introduce”.

**Questions:**

None

---

> ### Author Response · Authors · 2023-11-16
>
> We sincerely appreciate your valuable feedback on our work and for recognizing the contributions made by our research! We have diligently addressed the typos in the manuscript to enhance its quality. The revised content is distinctly highlighted in orange.

---

### Official Review · Reviewer_sHVZ · 2023-11-01

**Soundness:** 3 good
**Presentation:** 2 fair
**Contribution:** 3 good
**Rating:** 6
**Confidence:** 4

**Summary:**

This paper discovers practical limitations of conventional graph noise in terms of node features, i.e., the noise in node features is independent of the graph structure or node label. To mitigate the limitations of the existing assumption, the paper introduces a more realistic graph noise scenario called feature-dependent graph-noise (FDGN). Technically, the paper devises a deep generative model that directly captures the causal relationships among the variables in the DGP of FDGN and also derives a tractable and feasible learning objective based on variational inference. Empirically, the paper justifies the effectiveness of FDGN by conducting experiments on six datasets with both node classification and link prediction tasks.

**Strengths:**

The investigated problem of graph noise is essential. The paper breaks the existing assumption of feature noise, which is new to the community.

The paper is solid and extensive from a technical perspective.

The presentation and drawn figures are generally clear and easy to understand.

The paper is also theoretically grounded, with detailed justification elaborated on.

Several technical details, case studies, and evaluation results are also elaborated on in the Appendix.

**Weaknesses:**

Although some basic examples are given, the practical existence of causal relationships among X, A, and Y, i.e., "$A ← X, Y ← X, Y ← A$," should be further justified and supported by real-world evidence and materials. In other words, the paper should further explain why, in reality, noisy node features may entail both structure and label noise to be more convincing and practically worthy, especially in e-commerce systems.

Further, if "$A ← X, Y ← X, Y ← A$" is true, why does the paper not choose to directly learn a clean latent $Z_X$ but choose to learn two latent variables $Z_A, Z_Y$.

The overall novelty is neutral. The technical key contributions of the paper are within the proposed causal model and its instantiation with a variational inference network. It skillfully combines both worlds and designs a relatively complex objective based on the KL divergence.

The writing can be largely improved. For example, there are too many "i.e., A/X/Y" in Section 3.1, which do not provide any further information but simple notations.
Besides, I would suggest the paper analyze the complexity of FDGN and provide running time or training curves.

In addition, most of the references are before 2023. I would suggest the paper have a discussion with one work [1] using variation inference for causal learning and one work [2] learning latent variables $Z_A, Z_Y$ for structural denoising, which are technically relevant to the proposed FDGN.

[1] GraphDE: A Generative Framework for Debiased Learning and Out-of-Distribution Detection on Graphs. NeurIPS 2022.

[2] Combating Bilateral Edge Noise for Robust Link Prediction. NeurIPS 2023.

**Questions:**

Please refer to the above weakness part.

---

> ### Author Response · Authors · 2023-11-17
>
> **(W1) Additional real-world evidences of FDGN**
>
> Upon the reviewer’s request for additional application examples of FDGN, we provide additional examples of real-world applications:
>
> **Biological networks**
>
> A cell-cell graph is widely used in computational biology [1,2,3]. In the cell-cell graph, a node denotes a cell, each node is associated with features that represent gene expression values, and each node is labeled with the cell type. As the graph structural information is missing by nature, a cell-cell graph is usually constructed based on the node feature similarity [1,2,3]. However, gene expression values (i.e., cell-gene count matrix) often contain noise due to dropout phenomenon and batch effect, and such noise may entail noisy cell-cell graph structures, which can be referred to as FDGN. Furthermore, since the cell type (i.e., node label) is annotated by using transcripted marker genes (i.e., important features), noisy node features may lead to noisy node labels, which can be referred to as FDGN.
>
> [1] scGNN is a novel graph neural network framework for single-cell RNA-Seq analyses. Nature communications 2021
>
> [2] scGCL: an imputation method for scRNA-seq data based on graph contrastive learning. Bioinformatics 2023
>
> [3] Single-cell RNA-seq data imputation using Feature Propagation. ICML 2023 Workshop on Computational Biology
>
> **Recommender systems in many domains**
>
> In recommender systems of various domains (e.g., e-commerce, news, movie, etc), user-item interaction graphs are common, where node feature represents the information of users/items, and the user-item interaction represents the graph structures. Besides, users’ communities (interests) and items’ categories denote node labels. In this situation, if a user creates a fake or incomplete profile due to various reasons (e.g., privacy), node features would become noisy. Due to such noisy node features, items that are irrelevant to the user’s genuine interest can be exposed to users, which would make them interact with irrelevant items, resulting in noisy user-item interaction graph structures. This is an example of FDGN. Furthermore, such noisy information of users and noisy user-item interactions may also eventually change the users’ communities (i.e., noisy node labels), which can be referred to as FDGN.
>
> We appreciate the feedback and have included these examples in Appendix H. We believe it effectively addresses the concern.
>
> **(W2) Why not directly learn a clean latent $Z_X$**
>
> In the data generation process (DGP) we introduce, latent variable $Z_Y$ is a cause of $X$, as node features are generated based on the node label, similar to the approach in [1]. However, there might be some cases in which $Z_Y$ can not determine $X$. In such cases, we are open to exploring the reviewer's suggestion of positing a causal relationship $X ← Z_X$, where $Z_X$ represents the latent clean node features, rather than $X ← Z_Y$. However, the direct inference of $Z_X$ presents some challenges, as it would require parameterizing the clean node feature matrix $X \in R^{N \times F}$, which could be computationally intensive when dealing with large $N$ and $F$. Nevertheless, we appreciate the reviewer's suggestion and leave it as a future work. Exploring this idea further could potentially enhance the applicability of our proposed method.
>
> [1] Instance-Dependent Label-Noise Learning under Structural Causal Models, NeurIPS 2021
>
> **(W3) Technical contribution is neutral**
>
> We acknowledge the reviewer's perspective on the technical contribution of our work in model derivation. However, we would like to emphasize that our primary contribution lies in proposing the DGP of FDGN, which is a more realistic scenario in graph learning. Additionally, we effectively adapt existing techniques, such as variational inference, to address our problem of interest. It is important to note that even after deriving the objective using variational inference, how to implement each term in the derived objective was also non-trivial and challenging, which is another technical contribution. We specified the challenges in instantiating each term in [here](https://openreview.net/forum?id=kNGxg8shA1&noteId=bVLGnXwfMA) compared to [1] that also takes a similar approach to our work. Moreover, to the best of our knowledge, our proposed method successfully tackles a novel and crucial problem for the first time: handling noise scenarios where node features, graph structures, and node labels all simultaneously contain noise.
>
> We appreciate the reviewer for bringing up this important concern. We believe that it effectively addresses the concerns raised by the reviewer.
>
> [1] Instance-Dependent Label-Noise Learning under Structural Causal Models, NeurIPS 2021

---

> > ### Author Response · Authors · 2023-11-17
> >
> > **(W4-1) There are too many "i.e., A/X/Y" in Section 3.1**
> >
> > We apologize for any confusion caused by the repetition of notation. We will make the necessary corrections to improve the clarity of our writing.
> >
> > **(W4-2) Analyzing the complexity of FDGN**
> >
> > We concur with the reviewer's opinion regarding the potential high computational complexity of our proposed method.
> >
> > Hence, we compare the training time of PRINGLE with the baselines to analyze the computational complexity of PRINGLE. In the below table, we report the total training time and training time per epoch on Cora with FDGN 50% for all models. Note that since STABLE is a 2-stage method, we did not report the training time per epoch. The results show that PRINGLE requires significantly less total training time and training time per epoch compared with WSGNN, ProGNN, RSGNN, STABLE, NRGNN, and RTGNN. This suggests that PRINGLE can be efficiently trained, while still achieving substantial performance improvements.
> >
> > Although AirGNN and EvenNet require much less training time than PRINGLE, their node classification accuracy is notably worse than other methods, including PRINGLE. This indicates that, despite their fast training time, they may not be suitable for real-world scenarios. In summary, PRINGLE demonstrates superior performance compared to the baselines while maintaining acceptable training time.
> >
> > |                             | WSGNN  | AirGNN | ProGNN | RSGNN  | STABLE | EvenNet | NRGNN  | RTGNN  | PRINGLE |
> > |-----------------------------|--------|--------|--------|--------|--------|---------|--------|--------|---------|
> > | Total training time (sec)   | 93.90  | 20.9   | 702.14 | 159.87 | 53.33  | 0.81    | 100.33 | 118.76 | 46.27   |
> > | Training time / epoch (sec) | 0.19   | 0.04   | 1.77   | 0.16   | -      | 0.004   | 0.20   | 0.18   | 0.09    |
> >
> >
> > We appreciate the feedback and have incorporated these experimental results into Appendix J. We believe it effectively addresses the concerns related to computational complexity.
> >
> > **(W5) Discussions about some recent works**
> >
> > We thank the reviewer for suggesting the two papers [1,2] that are relevant to FDGN.
> >
> > [1] introduced a debiased learning framework, GraphDE, designed to handle situations where out-of-distribution (OOD) samples are present in training data. Given that a noisy sample can be viewed as an OOD sample, GraphDE shares a similar objective with our work. Additionally, GraphDE employs a generative model based on variational inference, relevant to learning approach. However, there are significant distinctions in the DGP that we assume compared to GraphDE. Specifically, GraphDE assumes that a latent variable $e$ determines whether an instance is an OOD sample or not, overlooking the way that the OOD sample is generated. Conversely, the DGP of our proposed FDGN characterizes the process by which noisy samples (i.e., OOD sample) are generated (e.g., in a feature-dependent manner). This crucial difference enables a more fine-grained model learning than GraphDE, which may weaken the applicability of GraphDE to complex real-world noise scenarios, such as FDGN.
> >
> > [2] introduced a graph structure denoising framework called RGIB, primarily centered on the link prediction task. In their work, while $Z_A$ and $A$ share the same meaning as in our paper, $Y$ denotes "edge labels" rather than node labels (i.e., it serves as a binary indicator for the presence of query edges), and $Z_Y$ refers to "clean edge labels" rather than clean latent node labels. These distinctions significantly differentiate their approach from ours.
> >
> > Furthermore, it's important to note that RGIB primarily addressed noisy graph structures while assuming that node features and node labels are noise-free. In contrast, our proposed method, PRINGLE, is developed under the more realistic FDGN assumption, where node features, graph structures, and node labels all simultaneously contain noise. This fundamental difference underscores the enhanced applicability and robustness of PRINGLE compared to RGIB.
> >
> > We appreciate the reviewer's suggestion to include more recent references, and we have taken this feedback into account. After a thorough review, we have incorporated the suggested two papers into Appendix K to ensure the coverage of recent developments in the field.
> >
> >
> > [1] GraphDE: A Generative Framework for Debiased Learning and Out-of-Distribution Detection on Graphs, NeurIPS 2022
> >
> > [2] Combating Bilateral Edge Noise for Robust Link Prediction, NeurIPS 2023

---

> > > ### Comment · Reviewer_sHVZ · 2023-11-23
> > >
> > > Thanks for providing the responses that have addressed most of my concerns. I would suggest the paper highlight the practical existence of such noise and improve the presentation as promised. I will retain my score and suggest an acceptance.

---

### Official Review · Reviewer_MMoz · 2023-11-01

**Soundness:** 3 good
**Presentation:** 2 fair
**Contribution:** 2 fair
**Rating:** 3
**Confidence:** 3

**Summary:**

The paper show that many existing robustness-enhancing methods assume noise in node features is independent of the graph structure or node labels. This is potentially an unrealistic assumption in real-world situations. In response, the authors propose a novel noise scenario called feature-dependent graph-noise (FDGN) and an accompanying generative model to address it.

**Strengths:**

1. The experiments are extensive.
2. The performance is good.
3. A new dataset is introduced.

**Weaknesses:**

1. The proposed setting is a combination of popular GNN with label noise and [1]. It is better to clarify more application examples in real-world.
2. A lot of GNN with label noise works are missed [2].
3. The abstract cannot summarize the methodology, which makes the paper unreadable.
4. Why the last three losses share the same weights in Eq. 4?
5. Why the generative methods can release the label noise?


[1] Towards Robust Graph Neural Networks for Noisy Graphs with Sparse Labels

[2] Learning on Graphs under Label Noise

**Questions:**

See above

---

> ### Author Response · Authors · 2023-11-16
>
> **(W1-1) FDGN is a combination of structure noise and label noise.**
>
> We would like to emphasize that our proposed FDGN is not a naive combination of independent node label noise and graph structure noise (which is referred to [1] by the reviewer). In this work, we introduce a more realistic noise scenario, where **noisy node features may entail both structure and label noise**, which we call feature-dependent graph-noise (FDGN). In other words, we assume that there exist "causal relationships" among node feature noise, graph structure noise, and node label noise, rather than assuming that these noise occur independently, lacking any causal connections between them. We argue that the proposed FDGN is more comprehensive and applicable in various real-world scenarios compared to the naive combination, because such causal relationships are evident in a range of real-world applications (as will be described below).
>
> **(W1-2) More application examples in the real-world of FDGN**
>
> Upon the reviewer’s request for additional application examples of FDGN, we provide additional examples of real-world applications:
>
> **Biological networks**
>
> A cell-cell graph is widely used in computational biology [1,2,3]. In the cell-cell graph, a node denotes a cell, each node is associated with features that represent gene expression values, and each node is labeled with the cell type. As the graph structural information is missing by nature, a cell-cell graph is usually constructed based on the node feature similarity [1,2,3]. However, gene expression values (i.e., cell-gene count matrix) often contain noise due to dropout phenomenon and batch effect, and such noise may entail noisy cell-cell graph structures, which can be referred to as FDGN. Furthermore, since the cell type (i.e., node label) is annotated by using transcripted marker genes (i.e., important features), noisy node features may lead to noisy node labels, which can be referred to as FDGN.
>
> [1] scGNN is a novel graph neural network framework for single-cell RNA-Seq analyses. Nature communications 2021
>
> [2] scGCL: an imputation method for scRNA-seq data based on graph contrastive learning. Bioinformatics 2023
>
> [3] Single-cell RNA-seq data imputation using Feature Propagation. ICML 2023 Workshop on Computational Biology
>
> **Recommender systems in many domains**
>
> In recommender systems of various domains (e.g., e-commerce, news, movie, etc), user-item interaction graphs are common, where node feature represents the information of users/items, and the user-item interaction represents the graph structures. Besides, users’ communities (interests) and items’ categories denote node labels. In this situation, if a user creates a fake or incomplete profile due to various reasons (e.g., privacy), node features would become noisy. Due to such noisy node features, items that are irrelevant to the user’s genuine interest can be exposed to users, which would make them interact with irrelevant items, resulting in noisy user-item interaction graph structures. This is an example of FDGN. Furthermore, such noisy information of users and noisy user-item interactions may also eventually change the users’ communities (i.e., noisy node labels), which can be referred to as FDGN.
>
> We appreciate the feedback and have included these examples in Appendix H. We believe it effectively addresses the concern.

---

> ### Author Response · Authors · 2023-11-16
>
> **(W2) A lot of GNN with label noise works are missed**
>
> We agree with the reviewer’s comment about the missing baselines. Hence, we further compare PRINGLE with the widely used label noise baselines: Co-teaching+ [1], CP [2], D-GNN [3], and CGNN [4].
>
> [1] How does disagreement help generalization against label corruption? ICML 2019
>
> [2] Adversarial Label-Flipping Attack and Defense for Graph Neural Networks. ICDM 2020.
>
> [3] Learning graph neural networks with noisy labels. ICLR LLD Workshop 2019
>
> [4] Learning on Graphs under Label Noise. ICASSP 2023
>
> From the below table, we clearly see that the proposed method, PRINGLE, outperforms all the baselines in terms of the node classification accuracy. We argue that these baseline methods are designed to tackle the noisy node labels while assuming that both node features and graph structures are noise-free. However, the proposed FDGN introduces a more realistic noise scenario, wherein node features, graph structures, and node labels simultaneously contain noise. As a consequence, the performance of these existing methods is considerably more constrained compared to PRINGLE.
>
> |  Dataset |  Setting  | Coteaching+ |    CP    |   D-GNN  |   CGNN   |  PRINGLE |
> |:--------:|:---------:|:-----------:|:--------:|:--------:|:--------:|:--------:|
> |   Cora   |   Clean   |   84.7±0.9  | 84.3±0.4 | 83.7±0.5 | 85.2±0.7 | **86.2±0.7** |
> |          | FDGN-10\% |   76.9±0.5  | 78.7±0.6 | 78.6±0.5 | 77.4±0.3 | **82.9±0.6** |
> |          | FDGN-30\% |   66.0±0.8  | 68.2±0.5 | 68.5±0.4 | 69.2±0.8 | **78.2±0.3** |
> |          | FDGN-50\% |   55.0±0.1  | 53.1±0.9 | 57.7±0.3 | 55.1±0.2 | **69.7±0.6** |
> | Citeseer |   Clean   |   72.7±0.4  | 72.8±0.9 | 75.1±0.2 | 71.1±0.9 | **77.3±0.6** |
> |          | FDGN-10\% |   67.7±0.6  | 68.4±0.8 | 69.0±0.5 | 65.6±0.4 | **74.3±0.9** |
> |          | FDGN-30\% |   55.0±0.9  | 56.8±0.9 | 54.0±0.7 | 54.1±0.3 | **65.6±0.6** |
> |          | FDGN-50\% |   47.4±0.8  | 46.5±1.0 | 44.7±0.1 | 46.9±1.4 | **59.0±1.8** |
> |   Photo  |   Clean   |   93.1±0.0  | 93.3±0.5 | 93.1±0.1 | 92.7±0.5 | **94.8±0.3** |
> |          | FDGN-10\% |   87.9±0.8  | 90.5±0.5 | 90.3±0.6 | 87.1±0.3 | **93.2±0.2** |
> |          | FDGN-30\% |   83.1±0.2  | 85.1±1.0 | 85.9±0.3 | 85.1±0.2 | **90.5±0.4** |
> |          | FDGN-50\% |   81.5±0.3  | 80.4±0.6 | 85.1±0.8 | 80.5±0.7 | **87.6±0.2** |
> |   Comp  |   Clean   |   88.6±0.8  | 90.7±0.3 | 89.4±0.9 | 90.0±0.5 | **92.2±0.0** |
> |          | FDGN-10\% |   85.6±0.6  | 87.1±0.8 | 86.8±0.4 | 83.0±0.4 | **89.8±0.2** |
> |          | FDGN-30\% |   81.5±0.3  | 82.8±0.6 | 82.9±0.5 | 82.3±0.8 | **86.9±0.3** |
> |          | FDGN-50\% |   72.8±0.9  | 74.3±1.0 | 74.5±0.6 | 75.3±0.1 | **82.2±0.4** |
>
> We acknowledge and appreciate the feedback provided, and as a response, we have incorporated the corresponding results in Appendix I. We are confident that this inclusion effectively addresses the raised concerns.
>
>
> **(W3) The abstract cannot summarize the methodology.**
>
> We apologize for the oversight in the abstract. We have revised the abstract to include a more comprehensive summary of the proposed method and have uploaded the revised manuscript. The revised content is distinctly highlighted in orange.
>
> **(W4) Why the last three losses share the same weights in Eq. 4?**
>
> In our experiments, we observed that the last three loss terms, namely $L_{rec-feat}$, $L_{cls-dec}$, and $L_{p}$, have a relatively minor impact on the model's performance compared to the others. As a result, we have made a strategic decision to simplify the hyperparameter search process and improve the practicality of PRINGLE by sharing the weights $\lambda_3$ among these last three loss terms.
>
> We appreciate the reviewer's feedback and have addressed this concern by including a detailed explanation in the implementation details section in Appendix D.5. The revised content is distinctly highlighted in orange.
>
> **(W5) Why the generative methods can release the label noise?**
>
> We acknowledge that deep generative models typically do not address label noise as a primary focus. However, in the context of our proposed FDGN, the DGP accounts for label noise that is generated in a feature-dependent manner. Hence, directly modeling the DGP of FDGN is beneficial for the model to recognize the way that the label noise is generated. Consequently, our modeling approach effectively mitigates the impact of the label noise.

---

> ### Author Response · Authors · 2023-11-21
>
> We kindly request the reviewer to refer to our response addressing the concerns related to the plausibility of FDGN, missing baselines, and so on. If the reviewer have any additional concerns, please don’t hesitate to bring them to our attention. We are eager to address any further concerns or questions the reviewer may have.

---

> ### Comment · Reviewer_MMoz · 2023-11-22
> **[Re] Rebuttal by Authors**
>
> Thanks for your efforts in the rebuttal. I have read all the reviews and responses. The authors added some missing baselines and revised the abstract, which improved the quality of the paper. However, the limited novelty of the proposed setting and techniques in this paper is still the main concern, which I haven't been persuaded. It makes the paper not achieve the requirements for ICLR. I would nonetheless take these comments into consideration in the reviewers' discussion and in the final score.

---

> ### Author Response · Authors · 2023-11-23
> **Final remarks to Reviewer MMoz**
>
> We thank the reviewer for carefully reviewing our rebuttal and other reviews. Finally, we would like to briefly underscore our contributions and respectfully request the reviewer to take them into consideration during the upcoming reviewers' discussion phase:
>
> 1. Our primary contribution lies in proposing the DGP of FDGN, which is well justified through a range of real-world application examples. Furthermore, the proposed FDGN is a new and novel concept in graph learning, which is a pivotal technical contribution.
> 2. We properly derive the ELBO objective and carefully instantiate each term in the objective while addressing various intricate challenges.
> 3. For rigorous evaluation, we newly introduce two real-world graph noise benchmark datasets, Auto and Garden, which is expected to foster practical research in noise-robust graph learning.

---

### Official Review · Reviewer_sE61 · 2023-11-01

**Soundness:** 2 fair
**Presentation:** 2 fair
**Contribution:** 2 fair
**Rating:** 5
**Confidence:** 4

**Summary:**

This paper proposed a new setting under graph weakly-supervised learning, named feature-dependent graph-noise, where the noise could be presented on either edge, label, and feature. To counter this proposed noise, authors leveraged the variational autoencoder (VAE) to model the latent variable and capture the causal relationship.

**Strengths:**

1. Authors adapt a causal prospective to justify the feature-dependent graph-noise, which is intuitive and sensible under mild assumptions.

2. This paper is overall well-presentated, the ideas are easy-to-follow.

3. The proposed metod demonstarates strong performances over multiple settings (graph noise, edge noise, label noise, feature noise).

**Weaknesses:**

1. The proposed solution lacks technical novelty, using VAE to model the causal relationship and counter noise has already been proposed by [1]. This paper only incrementally adapts that solution on the graph.

2. The proposed solutions lack theoretical support, the derivation on ELBO are well-known results, and the authors are only re-stating them here.

3. The proposed solution seems to have very high complexity (there are three encoder-decoder pairs, and three objectives to compute), therefore an efficiency analysis is needed.

**Questions:**

not at the moment

---

> ### Author Response · Authors · 2023-11-12
> **Citation clarification request**
>
> We sincerely thank the reviewer for thoughtful and constructive feedback on our paper. However, we kindly request the reviewer to provide clarification regarding the specific citation details for [1] mentioned in Weakness 1, which appears to be missing. This information would be invaluable for us to address the concern effectively.

---

> ### Comment · Reviewer_sE61 · 2023-11-12
> **Adding references to the review**
>
> Dear authors, I apologize for this oversight, and thank your prompt identification of the issue. The article I'm referring to is CausalNL proposal by Yao et al., which is in my view very similar in terms of scope and approach to this work. Notably, both studies explore the application of a causal perspective to address the challenge of learning in the presence of noise. Furthermore, both works leverage variational auto-encoders to introduce an additional reconstruction loss to model latent variables.
>
> Yao, Yu, et al. "Instance-dependent label-noise learning under a structural causal model." Advances in Neural Information Processing Systems 34 (2021): 4409-4420.

---

> ### Author Response · Authors · 2023-11-16
>
> **(W1) This paper only incrementally adapts CausalNL [1] on the graph.**
>
> We agree with the reviewer's opinion that our work may appear to lack technical novelty in comparison to [1]. However, it is important to note that we tackle additional, more challenging aspects unique to our specific problem that do not exist in [1]. Specifically, when additionally introducing $A$, we handle four more causal relationships: $\epsilon → A, X → A, A → Y, Z_A → A$, each of which is non-trivial to consider. We would like to specify the challenges in instantiations caused by their additional introduction.
>
> **Inference of $\epsilon$**
>
> In contrast to [1], which assumes that $\epsilon$ is a cause of $X$, our proposed FDGN consider that $\epsilon$ is a cause of both $X$ and $A$. In other words, the DGP of FDGN contains the causal relationships $\epsilon → X$ and $\epsilon → A$, as real-world applications often exhibit graph structure noise originating from arbitrary sources (i.e., $\epsilon$) in addition to the feature-dependent noise. Therefore, this scenario is a unique characteristic of our problem and not addressed in [1]. To deal with this, we decompose $q_{\phi_2}(\epsilon |X,A,Z_Y)$ into $q_{\phi_{21}}(\epsilon_X | X, Z_Y)$ and $q_{\phi_{22}}(\epsilon_A|X,A)$. While the instantiation of $q_{\phi_{21}}(\epsilon_X | X, Z_Y)$ is similar to [1], that of $q_{\phi_{22}}(\epsilon_A|X,A)$ is non-trivial and is absent in [1]. In our approach, we regard $\epsilon_A$ as a set of scores indicating the likelihood of each observed edge being noisy or not. Moreover, we leverage the concept of early-learning phenomenon to infer $\epsilon_A$. It is worth emphasizing once again that the instantiation of this scenario is novel and not a straightforward extension of [1].
>
> **Loss term $kl(q_{\phi_1}(Z_A|X,A)||p(Z_A))$**
>
> To compute this loss term, we encounter two primary challenges:
> 1. Designing an appropriate prior for the latent graph structure $p(Z_A)$
> 2. Addressing the complexity associated with calculating the KL divergence between the two matrices sampled from Bernoulli distributions.
>
> In response to the first challenge, we employ the $\gamma$-hop subgraph similarity as a metric to identify assortative edges.
> Regarding the second challenge, we introduce a predefined candidate graph, which includes the observed edge set along with a $k$-NN graph based on the $\gamma$-hop subgraph similarity. Both of these challenges represent non-trivial aspects of our approach and are absent in [1].
>
> **Loss term $kl(q_{\phi_3}(Z_Y|X,A)||p(Z_Y))$**
>
> To compute this loss term, [1] employs a uniform distribution as the prior $p(Z_Y)$. In contrast, we introduce the concept of class homophily to effectively regularize the inference of latent clean node label $Z_Y$. Specifically, we encourage $Z_Y$ to align with our prior knowledge, i.e., $p(Z_Y)$, that the two end nodes on the accurately inferred latent graph structure $Z_A$ are expected to have identical latent labels. Therefore, using this prior helps alleviate the noisy node label issues.
>
> It is essential to highlight that this instantiation effectively utilizes the property unique to the graph domain, which is not present in [1]. It is worth emphasizing once again that such an instantiation is novel and not a straightforward extension of [1].
>
> *To be continued in the following post*

---

> ### Author Response · Authors · 2023-11-16
>
> **Loss term $E_{Z_Y \sim q_{\phi_3}}[kl(q_{\phi_2}(\epsilon | X,A,Z_Y) || p(\epsilon))]$**
>
> This term is decomposed into $kl(q_{\phi_{21}}(\epsilon_X | X, Z_Y) || p(\epsilon_X))$ and $kl(q_{\phi_{22}}(\epsilon_A | X,A)||p(\epsilon_A))$. The first term is calculated in a similar manner to [1]. Specifically, a Gaussian distribution is employed as the prior $p(\epsilon_X)$. However, the second term is unique to our problem, and its computation necessitates prior information about $\epsilon_A$. As $\epsilon_A$ indicates the likelihood of each observed edge being noisy or not, it is not appropriate to simply assume $p(\epsilon_A)$ as a Gaussian distribution as done in [1]. This aspect makes the problem more challenging.
> To address this challenge, we introduce a new assumption that the inferred $\epsilon_A$ follows an unknown distribution with high variance, while our prior knowledge suggests that $p(\epsilon_A)$ follows the same distribution but with low variance. Additionally, we employ an Exponential Moving Average (EMA) technique to reduce the uncertainty of the inferred $\epsilon_A$. This challenge represent non-trivial aspects of our approach and is absent in [1].
>
> We appreciate the reviewer for bringing up this important concern. We have taken the feedback into consideration and have included an explanation in the Appendix G to further clarify this aspect of our work. We believe that it effectively addresses the concerns raised by the reviewer.
>
>
> [1] Instance-dependent Label-noise Learning under a Structural Causal Model, NeurIPS 2021

---

> ### Author Response · Authors · 2023-11-16
>
> **(W2) The derivation on ELBO are well-known results, and the authors are only re-stating them here.**
>
>
> We acknowledge that the derivation of the Evidence Lower Bound (ELBO) is based on well-known results. However, we would like to emphasize that our primary contribution lies in proposing the DGP of FDGN, which is a more realistic scenario in graph learning, not in the derivation of ELBO. In other words, considering that the ELBO of our objective has been derived by following the well-known derivation procedure, our main focus was on how to implement each term in the derived ELBO, which is non-trivial. Please refer to our response to [W1](https://openreview.net/forum?id=kNGxg8shA1&noteId=bVLGnXwfMA) for more detail on how each term is implemented. We fully agree with the reviewer’s opinion that the re-stating of the ELBO derivation may seem redundant as they are well-known, and we are willing to remove them from the paper.
>
> **(W3) An efficiency analysis is needed.**
>
> We thank the reviewer for pointing out the potential high computational complexity of our proposed method.
>
> Hence, we compare the training time of PRINGLE with the baselines to analyze the computational complexity of PRINGLE. In the below table, we report the total training time and training time per epoch on Cora with FDGN 50% for all models. Note that since STABLE is a 2-stage method, we did not report the training time per epoch. The results show that PRINGLE requires significantly less total training time and training time per epoch compared with WSGNN, ProGNN, RSGNN, STABLE, NRGNN, and RTGNN. This suggests that PRINGLE can be efficiently trained, while still achieving substantial performance improvements.
>
> Although AirGNN and EvenNet require much less training time than PRINGLE, their node classification accuracy is notably worse than other methods, including PRINGLE. This indicates that, despite their fast training time, they may not be suitable for real-world scenarios. In summary, PRINGLE demonstrates superior performance compared to the baselines while maintaining acceptable training time.
>
> |                             | WSGNN  | AirGNN | ProGNN | RSGNN  | STABLE | EvenNet | NRGNN  | RTGNN  | PRINGLE |
> |-----------------------------|--------|--------|--------|--------|--------|---------|--------|--------|---------|
> | Total training time (sec)   | 93.90  | 20.9   | 702.14 | 159.87 | 53.33  | 0.81    | 100.33 | 118.76 | 46.27   |
> | Training time / epoch (sec) | 0.19   | 0.04   | 1.77   | 0.16   | -      | 0.004   | 0.20   | 0.18   | 0.09    |
>
>
> We appreciate the feedback and have incorporated these experimental results into Appendix J. We believe it effectively addresses the concerns related to computational complexity.

---

> ### Author Response · Authors · 2023-11-21
>
> We kindly request the reviewer to refer to our response addressing the concerns related to the limited technical novelty and high complexity. If the reviewer have any additional concerns, please don’t hesitate to bring them to our attention. We are eager to address any further concerns or questions the reviewer may have.

---

> > ### Comment · Reviewer_sE61 · 2023-11-22
> > **Post-rebuttal summary**
> >
> > I appreciate the authors for their comprehensive rebuttal, especially their candor in acknowledging some of the potential drawbacks I mentioned.
> >
> > $\textbf{Summary}$
> >
> > 1. Overlapping methodlogy with CausalNL: this point has been agreed mutually that the detailed solution of this paper is largely overlapping with CausalNL. In addition, the series of points presented by the authors to highlight the non-trivial nature of addressing the problem under the context of robust graph learning has been noted. However, the following concerns persist:
> >
> > a. inference of $\epsilon$: this is an intuitive generalization of CausalNL on graph;
> >
> > b. reconstruction of $Z_{A}$: The challenges presented in this section do not appear sufficiently strong. There has been extensive existing works in this domain (edge re-construction), are challenges identified by the authors generally acknowledged as main challenges by other papers?
> >
> > c. reconstruction of $Z_{Y}$: The prior regularization seems post-hoc and herustic, also frequently used by other works;
> >
> > The arguments presented by the authors did not persuade me that the technical novelty in this paper is significantly enough.
> >
> > 2. Lack of theoritical guarantee: Despite the authors asserting that this is not their key contribution, the absence of theoretical support, especially under the context of overlapping methodology, is not favourable for the acceptance of this paper;
> >
> > 3. Efficiency analysis: this point has been addressed via experiments, surpursingly, the speed is comparable to most of the baseline methods, which I find adequate.
> >
> > Overall, I will maintain my score at the current stage, which I think is slightly below the acceptance threshold for a ICLR paper, however, during the discussion phase, I will not advocate for the rejection of this paper.
> >
> > Best of luck,
> >
> > reviewer sE61

---

### Meta-Review · Area_Chair_Z6Bs · 2023-12-08

**Metareview:**

This paper proposes a generative model designed to mitigate the side effects of noisy labels in graphs.

During the rebuttal, reviewers highlighted several common strengths. Specifically, 1) targeting an important problem; and 2) demonstrating good performance.

However, a critical concern is also raised by Reviewer sE61. Specifically, this paper appears to share overlapping techniques with another method for learning with noisy labels, CasualNL. This overlap significantly limits the technical contributions and insights provided by this paper.

The AC regretfully rejects it for now but encourages the authors to clearly demonstrate the originality and insights of this paper. We believe that this paper will be much stronger after addressing these issues.

**Justification For Why Not Higher Score:**

Technical contributions and insights provided by this paper may be limited. Specifically, this paper shares overlapping techniques with another method for learning with noisy labels, CasualNL. The major difference appears to be only in the assumed data generative process between the two papers.

**Justification For Why Not Lower Score:**

NA

---

### Decision · Program_Chairs · 2024-01-16

Reject